# Comparison of PMCAMx aerosol optical depth predictions over Europe with AERONET and MODIS measurements

**Antigoni Panagiotopoulou[1,2], Panagiotis Charalambidis[1,3], Christos Fountoukis[1], Christodoulos Pilinis[3], Spyros N. Pandis[1,2,4]**

[1]Institute of Chemical Engineering Sciences (ICE-HT/FORTH), Platani, P.O. Box 1414, Patras, 26504, Greece
[2]Department of Chemical Engineering, University of Patras, University Hill, Patras, 26504, Greece
[3]Department of Environment, University of the Aegean, University Hill, Mytilene, 81100, Greece
[4]Department of Chemical Engineering, Carnegie Mellon University, Pittsburgh, PA 15213, USA

*Correspondence to*: Spyros N. Pandis (spyros@chemeng.upatras.gr)

**Abstract.** The ability of the chemical transport model (CTM) PMCAMx to reproduce aerosol optical depth (AOD) measurements by the Aerosol Robotic Network (AERONET) and the Moderate Resolution Imaging Spectroradiometer (MODIS) over Europe during the photochemically active period of May 2008 (EUCAARI campaign) is evaluated. Periods with high dust or sea-salt levels are excluded so the analysis focuses on the ability of the model to simulate the mostly secondary aerosol and its interactions with water. PMCAMx reproduces the monthly mean MODIS and AERONET AOD values over the Iberian Peninsula, the British Isles, central Europe, and Russia with fractional bias less than 15% and fractional error less than 30%. However, the model overestimates the AOD over northern Europe most probably due to an overestimation of organic aerosol and sulfates. On the other end, PMCAMx underestimates the monthly mean MODIS AOD over the Balkans, the Mediterranean, and the South Atlantic. These errors appear to be related to an underestimation of sulfates. Sensitivity tests indicate that the evaluation results of the monthly mean AODs are quite sensitive to the relative humidity (RH) fields used by PMCAMx, but are not sensitive to the simulated size distribution and the black carbon mixing state. The screening of the satellite retrievals for periods with high dust (or coarse particles in general) concentrations as well as the combination of the MODIS and AERONET datasets leads to more robust conclusions about the ability of the model to simulate the secondary aerosol components that dominate the AOD during this period.

# 1 Introduction

Atmospheric aerosols are suspensions of solid and/or liquid particles in air that scatter and absorb light. The aerosol optical depth (AOD) is defined as the integrated extinction coefficient over the entire atmospheric column and is a measure of the total aerosol loading (King et al., 1999; Kokhanovsky, 2008; Vijayarachavan et al., 2008; Hidy et al., 2009). Calculations of AOD require knowledge of the aerosol vertical profile, including the particulate matter size distribution, chemical composition, and microphysical state (Seinfeld and Pandis, 2006).

Aerosol properties can be retrieved from ground-based measurements as well as from satellite earth observations (Holben et al., 1998; Levy et al., 2007a, b; Kokhanovsky, 2008; Levy et al., 2010; Duncan et al., 2014; Hu et al., 2014). Global observations of high spatial coverage are provided by satellites (King et al., 1999; Vijayarachavan et al., 2008; Hidy et al., 2009) and more limited spatial coverage by ground-based stations. Regarding temporal coverage, satellite observations are sparse when compared against ground measurements. Ground-based measurements of AOD are direct measurements while satellite AOD measurements are indirect, resulting from inversion procedures and exhibiting larger uncertainties. The magnitude of the satellite AOD uncertainties is higher over land where the surface reflectance cannot be neglected and it must be retrieved simultaneously with the aerosol properties (Levy et al., 2007a, 2007b, 2010). The satellite inversion procedure is simpler over water since the surface contribution is small and the detected signal is mostly due to aerosol reflectance (Shi et al., 2011; Anderson et al., 2013; Schutgens et al., 2013).

Chemical transport models (CTMs) are valuable tools for the study of the impact of pollutant emissions, the development of air quality improvement strategies, studies of aerosol radiative forcing, visibility, and global climate change. Uncertainties of the CTM's input data, including meteorological fields, emission inventories, and boundary conditions as well as weaknesses in representation of atmospheric processes may lead to weak model performance (Kinne et al., 2003, 2006). CTMs have been used in the past to provide AOD predictions either globally (Chin et al., 2002, 2004; Lee et al., 2010; Johnson et al., 2012; De Meij et al., 2012; Pozzer et al., 2012; Yu et al., 2012) or over specific regions like Asia (Han et al., 2010; Park et al., 2011), United States (Roy et al., 2007), and Europe (Jeuken et al., 2001; Hodzic et al., 2006; Meij et al., 2007; Tombette et al., 2008; Myhre et al., 2009; Carnevale et al., 2011; Im et al., 2014). Model evaluation often relies on in-situ ground measurements but also measurements from airborne platforms. These in-situ measurements cover by necessity a limited part of the modeling domain. Comparisons against remote sensing data have been used to close that gap.

Jeuken et al. (2001) compared the TM3 CTM AOD predictions with the ATSR-2 radiometer AOD retrievals during a 1997 summer episode over Europe. Model errors (neglecting organics and mineral aerosol) in the vertical distributions of sulfate, ammonium, and nitrate, in the hygroscopic growth, and in the optical parameters led to an average AOD (at 550 nm) underestimation by 0.17-0.19. Hodzic et al. (2006) used the CHIMERE model to simulate AOD at 865 nm over Europe during August 2003. The model generally reproduced AOD within a factor of 2 and with correlation coefficients ranging from 0.4 to 0.6 in comparison with POLDER and AERONET. Sporadic aerosol emissions due to forest fires or dust events led to regional AOD underestimations. De Meij et al. (2007) used the mesoscale TAPOM model to investigate AOD over Milan, Italy during June 2001. Simulated and observed AODs by AERONET, MODIS and MISR (Multi-angle Imaging Spectroradiometer) differed by a factor of 2 or 3 in days with cirrus clouds and Saharan dust but showed good agreement in clear sky days. A finer model resolution gave a more detailed AOD distribution pattern and improved by 15% the agreement with the AOD observations. Tombette et al. (2008) compared the Polair3D estimated AOD against AERONET measurements over Europe for 2001. The black carbon (BC) mixing state had almost no effect on the estimated single scattering albedo (SSA) but the aerosol water content influenced significantly both the SSA and the AOD. Myhre et al. (2009) used the global Oslo CTM2 to predict AODs at 550 nm focusing on specific European regions (Adriatic Sea, Black Sea, and Po Valley). Comparisons against AOD measurements from AERONET, MODIS, and MISR were presented for a short period during late summer-early autumn of 2004. The model underestimated AOD around Venice against AERONET because of organic carbon underestimation. Carnevale et al. (2011) implemented the TCAM CTM to simulate AODs during 2004 over Italy. In general, TCAM was found to underestimate MODIS AODs. Analysis of the extinction coefficient showed that the submicron inorganic aerosol played a key role. Im et al. (2014) simulated air pollution over Europe using the WRF-CMAQ modeling system for 2008. The model underestimated AERONET AOD measurements by 3-22% on average. AOD underestimations were attributed to underestimation of either the anthropogenic emissions or the natural and re-suspended dust emissions.

The $PM_1$ composition predictions of PMCAMx have been evaluated over Europe for the May 2008 EUCAARI intensive campaign (Fountoukis et al., 2011). The model performance was evaluated against ground measurements which were taken at stations located in the Netherlands, Greece, Ireland, and Germany as well as against airborne measurements from 15 flights in North-Western Europe. More than 94% of the organic aerosol (OA) hourly values and more than 82% of the sulfate ones were

reproduced within a factor of 2. PMCAMx performance against airborne measurements was as good as its performance against the hourly ground measurements.

One of the limitations of the previous AOD-based CTM evaluation exercises is that errors in dust emissions, transport, and removal often dominate the overall results. In the present work MODIS and AERONET AODs are filtered to exclude periods with high dust or sea-salt levels and to focus on the rest of the anthropogenic and biogenic aerosol components. A period with high photochemical activity is selected so that the emphasis is on secondary aerosol components. In this work we exclude for each locations periods characterized by high coarse particle concentrations, so $PM_1$ is the appropriate metric for aerosol composition evaluation.

In the present study we provide a first time evaluation of the ability of PMCAMx (Murphy and Pandis, 2009; Fountoukis et al., 2011) to reproduce AOD observations over Europe. The objective of this work is to identify weaknesses and strengths of PMCAMx and its inputs, by taking advantage of the wide spatial coverage of MODIS and the temporal coverage of AERONET. The major new methodological improvement in this effort is the screening of the satellite retrievals for periods with high dust (or coarse particles in general) concentrations as well as the combination of the MODIS and AERONET datasets. This combined with the high photochemical activity of the period allows us to focus on the ability of the model to predict secondary inorganic and organic aerosols and their interactions with water. The May 2008 period was chosen for two reasons. First it coincides with the EUCAARI campaign focusing on a photochemically active period with summertime-like conditions. Detailed continuous measurements of $PM_1$ composition both at the ground and aloft as well as a corresponding emission inventory (prepared by TNO) exist for that period. The second reason was that the ability of PMCAMx to reproduce these detailed $PM_1$ composition measurements has already been evaluated in previous work (Fountoukis et al., 2011; 2014) and therefore we can focus on the optical properties of the fine particulate matter in this paper. The exact dates simulated here were the same as in the previous publications for consistency.

## 2 PMCAMx description

PMCAMx is a three-dimensional CTM that employs the framework of CAMx (Environ, 2003) simulating the processes of horizontal and vertical advection, horizontal and vertical dispersion, wet and dry deposition as well as gas, aqueous, and aerosol-phase chemistry. Three detailed aerosol models are employed: inorganic aerosol growth (Gaydos et al., 2003; Koo et al., 2003), aqueous-phase chemistry

(Fahey and Pandis, 2001) as well as OA formation and chemical aging (Murphy and Pandis, 2009). The specific modules utilize a sectional approach that dynamically models the evolution of the aerosol size distribution. Ten size sections covering particle diameters from 40 nm to 40 μm are used. The model simulates the composition of each size section and therefore predicts the size-resolved PM composition using in this application 10 size bins. PMCAMx calculates the aerosol number from the corresponding mass distribution while its sister model, PMCAMx-UF, simulates both the aerosol number and mass distributions explicitly. Both primary and secondary organic PM is treated as semivolatile and photochemically reactive employing the volatility basis set (Murphy and Pandis, 2009). Additional details about the model can be found in Fountoukis et al. (2014), Tsimpidi et al. (2011), and Fountoukis et al. (2011).

The PMCAMx European modeling domain in this application is a region of 5,400 x 5,832 $km^2$ with 36x36 $km^2$ grid resolution and 14 vertical layers extending up to approximately 6 km. The considered period is May 2008 (EUCAARI campaign). Simulations were performed on a polar stereographic map projection. Horizontal wind components, vertical diffusivity, temperature, pressure, water vapor, clouds, and rainfall were provided by the Weather Research and Forecasting meteorological model (WRF) (Skamarock et al., 2008). We used hourly meteorological data from WRF as input to PMCAMx. WRF was driven by static geographical data as well as dynamic meteorological data (near real time and historical data that were generated by the Global Forecast System at $1 \times 1^o$). In the vertical dimension 27 sigma-p layers up to 0.1 bars were employed. Each PMCAMx layer is aligned with the WRF layers. WRF was periodically (every 3 days) reinitialized in order to increase the accuracy of the meteorological input fields to PMCAMx. Anthropogenic gas and inorganic aerosol emissions are from the GEMS European emissions database, while elemental carbon and organic carbon emissions are from the EUCAARI Pan European Carbonaceous Aerosol Inventory (Kulmala et al., 2011). This carbonaceous aerosol inventory was derived from the IIASA's GAINS inventory (Klimont et al., 2002; Kupiainen and Klimont, 2004) by application of source specific elemental carbon and organic carbon fractions. Details about the development of the EUCAARI TNO emissions can be found in Visschedijk et al. (2007) and Kulmala et al. (2011). Biogenic emissions were calculated by the MEGAN v2.04 model (Guenther et al., 2006). The marine aerosol emission model developed by O'Dowd et al. (2008) was employed for the estimation of mass fluxes for both accumulation and coarse mode, including the organic aerosol fraction. Emissions from wildfires are taken from IS4FIRES (Sofiev et al., 2009) and the size and composition distribution used is based on Andreae and Merlet (2001). Approximately 75% of the emissions are in the $PM_{2.5}$ fraction and the rest in the coarse fraction.

One baseline model simulation for May 2008 was performed together with a number of additional sensitivity tests described in subsequent sections. Given that the initial conditions are quite uncertain and dominate the model predictions during the first few days, we have excluded the corresponding "start-up" period (first six days) from the model evaluation. Concentrations of the major $PM_{2.5}$ aerosol components at the boundaries of the domain (Table S1 in Supplementary Information) are based on measurements of typical background concentrations in sites close to the domain boundaries (Zhang et al., 2007; Seinfeld and Pandis, 2006). All concentrations given here are under ambient temperature and pressure conditions.

## 2.1 AOD prediction by PMCAMx

The size and chemically resolved concentrations of aerosol particles are simulated by PMCAMx for every computational cell. Inorganic aerosol water concentration is calculated online by the thermodynamic equilibrium model ISORROPIA (Nenes et al., 1998). Taking into account all the vertical layers, we calculate the PMCAMx AOD at 550 nm as the sum of the extinction coefficients at each layer:

$$\text{AOD} = \sum_{i=1}^{14} b_{ext,i} \Delta z_i \tag{1}$$

where $b_{ext,i}$ is the extinction coefficient of layer $i$ and $\Delta z_i$ is the corresponding layer thickness. Assuming that the particles are homogeneous spheres and that all particles in each size bin have the same composition (internal mixture), the aerosol extinction coefficient ($b_{ext,i}$) for layer $i$ is:

$$b_{ext,i} = \sum_{j=1}^{10} \frac{\pi D_j^2}{4} N_j Q_{ext,j}\left(m_j, D_j\right) \tag{2}$$

where $D_j$ is the mean diameter of size bin $j$ and $Q_{ext,j}$ is the extinction efficiency of a single particle having a complex refractive index $m_j$. $N_j$ is the aerosol number concentration for bin $j$ calculated according to:

$$N_j = \frac{6c_j}{\pi p_j D_j^3} \tag{3}$$

where $c_j$, is total concentration of all aerosol chemical components and $p_j$ is the aerosol average density at size bin $j$. The extinction efficiency for bin $j$ is estimated as the sum of the scattering, $Q_{scat,j}$, and absorption, $Q_{abs,j}$ efficiencies:

$$Q_{ext,j} = Q_{scat,j} + Q_{abs,j} \tag{4}$$

Aerosol scattering and absorption efficiencies ($Q_{scat,j}$, $Q_{abs,j}$) are calculated using Mie theory (Seinfeld and Pandis, 2006) and mass concentrations provided for each size bin by PMCAMx, including the concentrations of particulate water. The mean bin diameter is used for the Mie computations. We have evaluated the accuracy of this simplification by performing detailed calculations over the full diameter range assuming uniform mass or number distributions in each bin. In all cases examined, the differences in the estimated AOD were at most a few percent justifying our simplification. The complex refractive, $m_j$, index of a homogeneous sphere is estimated using the volume weighted average of the individual refractive indices (Pilinis and Pandis, 1995). Sulfate and ammonium are assumed to have a real refractive index of 1.53, which is the value of ammonium sulfate (GEISA, 2011; NASA, 2006). Nitrate is assumed to have a real refractive index of 1.56, similar to the value of ammonium nitrate (NASA, 2006). Sodium and chloride have a real refractive index of 1.5 (GEISA, 2011; NASA, 2006). Dust is assumed to have a complex refractive index of 1.53-0.0055i (GEISA, 2011). OA is assumed to be non-absorbing with a refractive index of 1.5 (Nessler et al., 2005; Fierz Schmidhauser et al., 2010). Biomass burning was minimal during the period of interest (Crippa et al., 2014), so this simplifying assumption regarding the OA absorptivity has little effect on the predicted AOD. The black carbon refractive index has the largest uncertainties (Bond and Bergstrom, 2005) and we use a value of 1.75-0.44i (GEISA, 2011). In the base case BC is assumed to be internally mixed with the other components in each size range. The sensitivity of the model predictions to this assumption is discussed in a subsequent section. Biomass burning emissions in Europe during the simulated period were low and therefore any effects from biomass burning related brown carbon are also expected to be small.

## 3 MODIS and AERONET data

The cloud screened and quality assured Level 2 AERONET direct AOD measurements are used for the PMCAMx evaluation. AERONET applies the Beer Lambert Bouguer law to measure AOD from direct sun observations (Holben et al., 1998) therefore it is considered to be the ground truth with AOD uncertainties of 0.01 – 0.02 (Eck et al., 1999). The AERONET measurements have a variable temporal resolution which is on average around 15 min. Measurements start after sunrise when the sun is approximately 7.5 degrees above the horizon and end a little before sunset when the sun is once more at

approximately 7.5 degrees. We use here the AERONET AOD at 550 nm. In this work only the AOD values corresponding to Angstrom Exponent values greater than 0.9 are employed in an effort to exclude periods with high dust or high sea-salt levels (Schuster et al., 2006). This filter rejected 29% of AODs over land and 28% over water. The geographical distribution of the corresponding AERONET stations is depicted in Fig. 1 and the number of stations in each region is shown in Table 1. Some AERONET stations in the domain of interest did not have available Level 2 AOD data for the period of interest while all data from three stations (OHP_OBSERVATOIRE in South France, FORTH_CRETE in Crete, Greece, and ATHENS_NOA in Athens, Greece) have been excluded after the coarse particle (dust or sea-salt) rejection filtering.

The polar-orbiting MODIS monitors global aerosol properties from two satellites: Terra and Aqua (Salomonson et al., 1989). MODIS employs 36 channels from 0.412 to 14.2 μm, has a wide swath of 2,330 km, and observes every location of the globe at least once daily. The default resolution for aerosol retrieval is 10x10 km$^2$ (Levy et al., 2009). Each data set retrieved by MODIS is associated with a Quality Assurance Confidence (QAC) flag which ranges from 0 (no confidence) to 3 (highest confidence). For increased spatial coverage we use both the Terra and Aqua MODIS AOD retrievals with QAC $\geq$ 1. We employ the MODIS Level 2 Collection 5.1 aerosol datasets. The Dark-Target algorithm products were used. We did not alter the values of the data records and we did not apply any additional transformations. The MODIS AOD values, retrieved with spatial resolution 10x10 km$^2$, were assigned to the corresponding computational cells of the PMCAMx modeling domain. AOD retrievals are provided at seven wavelengths (470, 550, 660, 870, 1,200, 1,600, 2,100 nm) over water surface and four wavelengths (470, 550, 660, 2,100 nm) over land. In this study we focus on the 550 nm values. Figure 2 presents the geographical distribution of the available MODIS AOD measurements during the period of interest 1-29 May 2008 (EUCAARI campaign) over Europe. The average number of retrievals is 12±9. The maximum number of retrievals is 65 in areas in the North Atlantic.

Dust emissions from the Sahara are not included in the PMCAMx emissions used here and the focus of this study is on periods and regions in which Saharan dust or other coarse particles like sea-salt do not contribute significantly to the AOD. To exclude periods with high coarse particle levels and to focus on the rest of the anthropogenic and biogenic aerosol components, MODIS AODs are filtered. Over water we employ the coarse particle rejection filter of Barnaba and Gobbi (2004). According to this filter, AOD values greater than 0.3 also corresponding to coarse mode fraction higher than 0.3 are assumed to be coarse particle-influenced periods. Over land we only use the AOD values which correspond to Angstrom Exponent values exceeding 0.9 (Schuster et al., 2006). The above filters

discard 16% of MODIS AOD values over land and 0.4% over water. This is mostly due to the fact that a lot of the periods with high dust levels are also accompanied by cloud cover over water. As a result there are no MODIS AOD retrievals during these periods thus lowering the corresponding fraction that needs to be discarded due to dust influence. The location of the AERONET stations also contributes to this difference.

The evaluation of the MODIS AODs at 550 nm for the land algorithm was performed following the approach of Remer et al. (2005) and Levy et al. (2007b). Fig. S1 presents a comparison of the corresponding AERONET observations with the MODIS AOD retrievals AERONET measurements were spatially and temporally collocated with MODIS retrievals, similar to the scheme proposed by Ichoku et al. (2002). The collocated data were sorted according to the AERONET AOD observations. The resulting data were partitioned into groups of 100 AOD points and then averaged. At higher optical depths since the data became sparser we used 25 points for each bin. The regression line of the collocated AODs, prior to partitioning, had a slope of 1.05. 73% of the 8,331 collocated points fall within the expected error envelope. These results indicate that the mean MODIS AOD over land in the region and period of interest was retrieved with the expected accuracy. The highest quality flag QAC = 3 provides the closest match, but including the QAC = 2 and 1 retrievals results in only a minor reduction of accuracy while increasing significantly the size of the dataset (Table S2).

Previous studies have shown that MODIS AOD retrievals have an expected error of $\pm(0.05 + 0.15AOD_{AERONET})$ over land and $\pm(0.03 + 0.05AOD_{AERONET})$ over water (Chu et al., 2002; Remer et al., 2005; Levy et al., 2007a,b, 2010; Anderson et al., 2013). Table S3 summarizes the values of the expected MODIS AOD uncertainties for the various regions in our modeling domain during May 2008, based on the monthly mean values of AERONET AOD. The MODIS-AERONET AOD differences for this period are consistent with the expected uncertainty of the MODIS retrievals (Fig. S1).

## 4 Evaluation of PMCAMx fine PM composition and mass predictions

Fountoukis et al. (2011) evaluated the ability of PMCAMx to simulate the chemical composition of $PM_1$ components during the same period simulated in this study (May 2008) using the measurements of the intensive campaign of European Aerosol Cloud Climate and Air Quality Interactions (EUCAARI) project (Kulmala et al., 2011). The model predictions were compared with hourly averaged AMS ground measurements as well as airborne measurements over Europe (Morgan et al., 2010). The measurements covered Central Europe, England and Ireland, North Atlantic and the

Mediterranean. Approximately 8500 measurements (data points) from four ground stations and sixteen flights were used in this evaluation.

PMCAMx predictions were in close agreement with the AMS measurements at Cabauw for all species. The predicted monthly average concentrations for OA, nitrate, sulfate and ammonium were 4.0, 3.2, 2.2 and 1.9 $\mu g\ m^{-3}$ respectively compared to the measured average of 4.1, 2.5, 1.5 and 1.7 $\mu g\ m^{-3}$. The model reproduced approximately 90% of the hourly $PM_1$ OA data within a factor of 2. At Finokalia the average predicted concentration was 2.1 $\mu g\ m^{-3}$ for OA, 4.7 $\mu g\ m^{-3}$ for sulfate, 0.09 $\mu g\ m^{-3}$ for nitrate and 1.3 $\mu g\ m^{-3}$ for ammonium in comparison with the AMS measurements of 2.5, 5.2, 0.08, and 1.5 $\mu g\ m^{-3}$, respectively. At Mace Head PMCAMx reproduced 79% and 74% of the hourly $PM_1$ OA and sulfate hourly measurements within a factor of 2. However, greater errors were seen for $PM_1$ nitrate and ammonium, because of the bulk equilibrium assumption used in that PMCAMx application. In Melpitz the model reproduced more than 80% of the hourly $PM_1$ OA data within a factor of 2. Overall, PMCAMx agreement with the AMS ground measurements for all stations was encouraging. More than 70% of the hourly data points for $PM_1$ sulfate and 87% for $PM_1$ OA lay within the 2:1 and 1:2 error lines. As expected, the model performance based on daily averaged values was even better, reproducing 94% and 82% of the hourly data within a factor of 2 for OA and sulfate, respectively. Overall the model fractional bias for the ground stations was -0.1 for OA, 0.1 for sulfate, 0.2 for ammonium and 0.4 for nitrate. For the airborne measurements, the PMCAMx fractional bias was -0.2 for OA, 0.2 for sulfate, -0.3 for nitrate and -0.08 for ammonium.

PMCAMx predictions of the vertical distribution of sub-micron aerosol chemical composition were evaluated against the airborne AMS data. Both PMCAMx and LONGREX airborne observations showed low OA concentrations in the 2-6 km altitude range over Europe during the simulation period. The ability of the model to reproduce the high time resolution airborne measurements at various altitudes and locations was similar to its ability to simulate the ground level concentrations. PMCAMx reproduced almost 70% of the sulfate and OA concentrations within a factor of 2. For measured sulfate and OA higher than 1 $\mu g\ m^{-3}$, the model reproduced 77% and 75% of the corresponding measurements, respectively, within a factor of 2.

A detailed evaluation of the ability of PMCAMx to reproduce observations of the organic aerosol composition for the same May 2008 period has been presented by Fountoukis et al. (2014). The PMCAMx predictions using the Volatility Basis Set approach were compared against AMS positive matrix factorization results. The model correctly predicted the low concentrations of fresh primary transportation-related OA (<0.3 $\mu g\ m^{-3}$) at Melpitz and Finokalia. At Mace Head it showed a small

tendency towards underprediction of the same component with a mean error of -0.25 μg m$^{-3}$. Overall, in

the comparison of PMCAMx against AMS hydrocarbon-like OA (HOA) measurements from all

stations, the mean error was 0.26 μg m$^{-3}$ while the mean bias was less than 1 μg m$^{-3}$. Regarding

oxygenated OA (the major component of OA according to the measurements in all stations) the model

reproduced 83% of the measured values within a factor of 2. The model biases for organics and sulfate,

the two major PM$_1$ components, were quite similar at the ground and higher altitudes (Fountoukis et al.,

2011). For example for organic aerosol the mean bias was -0.4 μg m$^{-3}$ in both cases while for sulfate it

was +0.1 μg m$^{-3}$ at the ground and -0.1 μg m$^{-3}$ aloft. The model reproduced well the almost zero ground

PM$_1$ nitrate levels in the Eastern Mediterranean (Finokalia) (mean bias 0.02 μg m$^{-3}$) and the moderate

levels in Central Europe (Melpitz) (mean bias -0.1 μg m$^{-3}$).  In the high ammonium nitrate region based

on the Cabauw measurements the nitrate bias at the ground was +0.8 μg m$^{-3}$. The mean bias at higher

altitudes was -0.2 μg m$^{-3}$.

The predictions of PM$_{2.5}$ and PM$_1$ of PMCAMx can also be compared against the corresponding

EMEP daily average mass concentration measurements. A total of 795 data points in 26 ground stations

are available during the simulated period and have been used for the evaluation. The model showed

very little bias (fractional bias equal to -0.07) and reasonable scatter (fractional error equal to 0.49). The

average predicted PM$_{2.5}$ concentration was 9.07 μg m$^{-3}$ while the average observed was 9.82 μg m$^{-3}$.

This performance is quite similar to the one reported by Fountoukis et al. (2011) for the EUCAARI

stations and airborne campaign for the same period. Details about this intercomparison can be found in

the Supplementary Information.

## 24   5 Evaluation of PMCAMx AOD predictions

The coarse particle-screened monthly mean AODs for Europe during May 2008 retrieved by

MODIS and predicted by PMCAMx are shown in Fig. 3. The PMCAMx AODs have been calculated

for exactly the same periods as the MODIS retrievals to allow the direct comparison of the two. The

comparisons with the MODIS AOD retrievals correspond exactly in space and time, so the times

coincide with the satellites' overpasses.

The MODIS retrievals show high AOD values (> 0.25) over England, South Ireland, North Italy,

South Poland, East Romania, Greece, and North Atlantic. Low AOD values (< 0.1) were retrieved over

East France, Belgium, Sweden, and North Russia. PMCAMx predicts high AODs over England, South

Ireland, North Italy, and central Atlantic and low AODs over North Sweden, East Russia, North and

South Atlantic. The data sample size is small over North Africa due to the high levels of dust in these areas during the whole simulation period. As a result the corresponding coarse particle-screened AOD comparisons provide little information about the ability of PMCAMx to simulate fine PM in this region.

**5.1 Overall evaluation**

The difference between PMCAMx and MODIS monthly mean AODs is depicted in Fig. 4. PMCAMx AODs are higher than those of MODIS over England, Ireland, France, Germany, central and South Italy, North and East Europe, central, North and West Russia, West Balkans, and central Atlantic. On the other hand PMCAMx predicts lower AODs than MODIS over parts of Russia, North Italy, central and South Balkans, South Poland, North and South Atlantic, and the African coast of the Mediterranean. On a domain average basis PMCAMx predicts an AOD equal to 0.14 while MODIS retrieved 0.16. Detailed comparisons for each region can be found in Table 2. 94% of the monthly mean AOD values fall inside the expected MODIS error envelope over land (Fig. 5a). Over the whole domain the PMCAMx monthly mean AODs have a mean error of 0.05 and a fractional bias of -16% compared to the MODIS monthly mean AODs (Tables 2 and S4). The correlation coefficient $R$ between the MODIS monthly-average AODs and the PMCAMx predicted AODs was 0.51.

PMCAMx AODs were also compared with the AERONET values for the simulation period. Once more the comparisons were done for the grid cells of the AERONET stations and corresponding measurement periods. The PMCAMx monthly mean AODs had a mean error of 0.03 and a fractional bias of 4% compared to the AERONET monthly mean AODs (Table 1). The comparison of the PMCAMx with AERONET monthly mean AODs is summarized in Fig. 5b for the 50 AERONET stations which are employed in the present study. The correlation coefficient $R$ between the AERONET monthly-average AODs and the PMCAMx predicted AODs was 0.57.

**5.2 Regional evaluation**

The performance of the model for AOD combined with its performance for composition in the sites where there are ground and airborne PM composition measurements, can be used to reach some tentative conclusions about its performance in reproducing the fine PM levels and composition. These are clearly limited to the components dominating the AOD in each area and either suggest problems or lack of major errors. These are discussed for each region below. We adopt here the four levels of model performance proposed by Morris et al. (2005) to evaluate PM models based on their fractional bias and error. These levels vary from "excellent" (absolute fractional bias $\leq$ 15% and absolute fractional error $\leq$

35%) to "good" (absolute fractional bias ≤ 30% and absolute fractional error ≤ 50%) to "average" (absolute fractional bias ≤ 60% and absolute fractional error ≤ 75%) to "problematic" (absolute fractional bias ≥ 60% and absolute fractional error ≥ 75%).

*Spain and Portugal*: The relatively low AOD levels (0.11 for the 8 AERONET stations and 0.14 for MODIS) are reproduced well by PMCAMx (0.12 for the AERONET sites and 0.12 for the periods of the MODIS retrievals). The monthly mean PMCAMx AOD predictions have a mean error of 0.02 (AERONET) and 0.04 (MODIS) (Tables 1 and 2). The model shows little bias (5%) compared to the AERONET stations and a small tendency towards underprediction (-15%) compared to MODIS. 83% of the monthly mean PMCAMx AODs are within the expected MODIS error envelope. These results are consistent with the evaluation of PMCAMx against the daily ground $PM_{2.5}$ mass measurements in eight stations in Spain (Niembro, Campisabalos, Cabo de Creus, Barcarotta, Zarra, Penausende, Els Torms, O Savinao) in which the fractional bias of the model is -0.04 and the fractional error is 0.52 (Table S6). Sulfate and organic aerosol are the predicted major components of dry fine PM in Spain and Portugal (Table 3).

*Russia, Belarus, and Ukraine*: PMCAMx reproduces well (0.14 predicted versus 0.15 measured) the average AOD observations at the 5 AERONET stations in this region (2 in West Russia, 1 in Belarus, 1 in Ukraine, and 1 in Crimea) (Table 1). The model has a similar good performance against the MODIS retrievals (0.12 predicted versus 0.13 retrieved) (Table 2). As a result, the monthly mean PMCAMx AOD predictions have a low mean error of 0.02 (AERONET) and 0.04 (MODIS). PMCAMx shows a slight tendency towards underprediction (-11%) compared to AERONET and no bias (<1%) compared to MODIS. 92% of the monthly mean PMCAMx AODs are within the expected MODIS error envelope. Sulfates and organic aerosol are predicted to predominate in this region and it appears that PMCAMx performs reasonably well in this ground-level measurement poor region. Significant discrepancies between predicted and observed AOD over Russia were expected given the uncertainty in the corresponding emissions. However, the agreement was quite good with both AERONET and MODIS. This rather surprising result clearly requires additional investigation and could be due to offsetting errors.

*United Kingdom (UK) and Ireland*: This area was relatively polluted during the simulation period with high levels of nitrates, sulfates, and organic aerosol based on both PMCAMx predictions and the airborne measurements (Table 3). PMCAMx reproduces the relatively high average MODIS (0.23

predicted versus 0.21 retrieved) and AERONET (0.24 predicted versus 0.25 measured in the station of

Chibolton). The monthly mean PMCAMx AODs have a mean error of 0.04 compared to MODIS with a

small tendency towards overprediction (14%). 90% of the monthly mean PMCAMx AODs fall within

the expected MODIS error envelope. The encouraging agreement of PMCAMx retrievals over this

region is consistent with its good performance when compared against the EUCAARI airborne and

ground measurements in this region (Fountoukis et al., 2011; 2014) for the major fine PM components.

*Balkans*: The Balkans according to PMCAMx had some of the highest sulfate levels in the domain

during the simulation period (Table 3). The model underpredicts the AOD both against MODIS (0.14

predicted versus 0.19 retrieved) and the two AERONET stations (0.15 predicted versus 0.21 measured).

The corresponding fractional biases are -24% against MODIS and -33% against AERONET. However,

80% of the monthly mean PMCAMx AODs fall within the expected MODIS error envelope. These

results are consistent with the $PM_{2.5}$ mass concentration underprediction (fractional bias -24%) in the

station of Iskrba in Slovenia (Table S6). Given that most of the predicted AOD is due to the sulfate

these results suggest that the PMCAMx underprediction is probably due to their underestimation.

*Central Europe*: PMCAMx showed a small tendency towards overprediction of the moderate AODs in

this region compared to both AERONET (12%) and MODIS (13%). For example, overpredictions were

evident over France and Germany (Fig. 4). The corresponding fractional errors on a monthly average

basis were 22% against AERONET and 30% against MODIS. Organic aerosol, sulfate, and nitrate were

the major predicted fine PM components in central Europe during this period consistent with the

measurements in Melpitz and Cabauw (Fountoukis et al., 2011). PMCAMx overpredicted sulfate levels

(fractional bias equal to 0.3) and nitrate levels (fractional bias of 0.3) in Cabauw, but showed little bias

(equal to 0.01) for OA. These results appear consistent with the AOD overpredictions. However, in

Melpitz the model slightly underpredicted sulfate (fractional bias= -0.1) and underpredicted OA

(fractional bias= -0.3). Using the airborne measurements over this region, sulfate was overpredicted

(fractional bias of 0.2) and OA was underpredicted (fractional bias of -0.2). These results are also

consistent with the ground $PM_{2.5}$ mass concentration measurements in six stations in the area (Illmitz,

Payerne, Rigi, Weldoff, Schaunisland, and Ispra) in which the absolute fractional biases are less than

35% and the fractional errors less than approximately 50% (Table 6). There higher $PM_{2.5}$ errors in the

site of Montelibretti in Rome, but these are mostly due to the coarse resolution of the model. These

results suggest that the sulfate overpredictions can probably explain to some extent the AOD

overpredictions. Errors in the relatively humidity fields could also explain parts of these AOD discrepancies.

*East Europe*: PMCAMx slightly overpredicted the AODs in this region compared to both AERONET (24%) and MODIS (25%). 82% of the monthly mean PMCAMx AODs fell within the expected MODIS error envelope. PMCAMx predicts more frequently AODs > 0.1 than measured by AERONET during the corresponding period of measurements probably because of an overestimation of sulfates and organic aerosol. The ground $PM_{2.5}$ measurements in the two sites in Latvia (Rucava and Zoseni) are the highest in the domain and are seriously underpredicted by PMCAMx (Table S6) and inconsistent with the low to moderate MODIS AODs in the same area (Figure 3). The reasons for these high fine PM measurements are not clear and need additional investigation.

*North Europe*: PMCAMx reproduces the low pollution levels in this area with a mean AOD of 0.12 compared to 0.08 by the 4 AERONET stations. The absolute monthly mean errors are low: 0.04 (AERONET) and 0.06 (MODIS). However, there is significant fractional positive bias compared to both AERONET (36%) and MODIS (47%). 54% of the monthly mean PMCAMx AODs fall outside the expected MODIS error envelope. 53% of the hourly PMCAMx AODs are greater than 0.1 while only 18% of the AERONET values are greater than 0.1. A similar overprediction of the $PM_{2.5}$ concentrations can be seen in several stations in the area (Asprveten, Hyytiala, Lille Valby) while in some other stations (Rao and Vavihill) there is little or no overpredition (Table S6). Sulfates and organic aerosol are the dominant predicted fine PM components in this region and the model probably overestimates at least one of them.

*Turkey and Northern Africa*: There are only two AERONET stations in this area and PMCAMx underpredicts by 30% the corresponding moderate AOD measurements. However, the model performance appears to be much better against the MODIS retrievals covering a much bigger area and the underprediction drops to 13%. 79% of the PMCAMx AODs fall inside the expected MODIS error envelope. Sulfates and organic aerosol are the major fine PM components according to PMCAMx in these regions and they are probably slightly underestimated by the model.

*Mediterranean Sea*: PMCAMx exhibits a tendency towards underprediction (-24%) against MODIS and 58% of the monthly mean PMCAMx AODs fall inside the expected MODIS error envelope. The major discrepancies are evident in the southern part of the Mediterranean especially close to the African coast.

These suggest that dust may be partially responsible for the errors even after the filtering of the data. The model performance is better in the eastern Mediterranean (Fig. 4). Sulfates dominated the AOD in the Mediterranean during the simulation period according to PMCAMx. Both the organic aerosol and the sulfate were underpredicted in the site of Finokalia in Crete with a mean bias of approximately -0.5 μg m$^{-3}$ (Fountoukis et al., 2011). However, the concentration of sulfate in Finokalia was more than twice that of organic aerosol. Given the difference in hygroscopicity between the two, most of the AOD (more than 80%) is due according to PMCAMx to sulfates. The omission of the high coarse PM periods from the evaluation data set has also eliminated the high sea-salt concentration periods from areas over water. As a result, the present work offers little insight about the ability (or lack there-of) of PMCAMx to model sea salt in the various marine environments examined here.

*South Atlantic*: The PMCAMx AOD predictions are significantly lower (-45%) compared to the MODIS retrievals in this region with 74% of the monthly mean PMCAMx AODs falling outside the expected MODIS error envelope. Sulfate and sea-salt dominated the predicted AOD in this region in May 2008 and there is evidence that they may be underpredicted. However, errors in relative humidity or cloud contamination could be also responsible for these discrepancies (Anderson et al., 2013). The predicted concentrations over the western Atlantic are heavily influenced by the boundary conditions used for the left side of our modeling domain. Any underestimation of these boundary conditions could also explain the underestimation of the AOD in this area.

*North Atlantic*: The model performance is much better in the North than in the South Atlantic. The mean AOD error is 0.04 compared to MODIS with a tendency towards underprediction (-21%). 53% of the monthly mean PMCAMx AODs fall inside the expected MODIS error envelope. There is one AERONET station in this area (in Helgoland around 50 km from the coast of Germany) and PMCAMx predicts an average AOD equal to 0.16 compared to the 0.11 measured. Sulfates, organic aerosol, and sea-salt were the major predicted fine PM components in North Atlantic during May 2008 (Table 3). PMCAMx performed relatively well (absolute fractional bias less than 0.2 for both sulfate and OA) when compared with the EUCAARI airborne measurements in this region.

*Black Sea*: PMCAMx exhibits a tendency towards underprediction (-18%) versus MODIS in this relatively polluted region. 66% of the PMCAMx AODs fall within the expected MODIS error envelope. Sulfates were the major predicted fine PM component in the Black Sea during the simulation period.

The results of the PMCAMx-MODIS comparison for the various regions are summarized in Fig. 6. These results suggest that the variability of the MODIS retrievals exceeds that of the PMCAMx predictions for almost all areas. Based on the monthly average AERONET observations (see Figure 5b) there is no indication that the model underestimates the high AODs and overestimates the low ones. On average, it does a reasonable job in both. However, when one examines the individual measurements (Figure 6) the range of the measurements exceeds that of the predictions. There are a number of possible reasons for this behavior that is often encountered in chemical transport models. The use of the same anthropogenic emissions inventory every day (with the exception of weekends) is one reason. These emissions do vary from day to day, however the model uses their average missing in the process both ends of the actual air pollution distribution. Measurement uncertainty is a second reason. This will also tend to extend the range of the measured AOD distribution compared to the predicted one. Errors in meteorological fields can also contribute to this discrepancy. Other potential contributors to this discrepancy is the relatively coarse spatial resolution of the model inputs, missing short-term air pollution sources in the inventory, potential cloud contamination of the retrievals, etc.

The AERONET measurements provide an additional opportunity to test the ability of the model to reproduce the observed average diurnal AOD variation, at least for the approximately 12 hours for which measurements are available. Some of these comparisons are shown in Figures S3-S8 in the Supplementary Material. Overall, for over 90% of the hourly averaged AOD measurements the PMCAMx error was less than 50%, indicating that the agreement of the average AODs was not due to offsetting temporal errors.

## 6 Sensitivity analysis of the predicted AODs

There are various possible sources of bias in the PMCAMx predictions of AOD other than the concentration and composition of aerosol. We explore here the role of the relative humidity calculated by the WRF model, the role of the mixing state of BC, as well as that of the predicted aerosol size distribution.

In the first test the absolute humidity was increased uniformly by 5%, while maintaining the maximum relative humidity in cloud-free regions at 99%. The PMCAMx monthly mean AOD increased on average by 13% (Fig. S2). The increases ranged from 7% in Turkey and Northern Africa to 31% in the North Atlantic. This AOD change can explain a significant part of the base case discrepancies which cause a fractional error of PMCAMx 22% versus AERONET and 33% versus MODIS.

In another test the diameter of all particles was increased by 20% without keeping the particle number constant. 72% of the PMCAMx monthly mean AOD values changed by less than 0.01. The average increase of the monthly mean AOD was 1% (ranging from 0.3% in the Black Sea to 4% in the UK and Ireland). This small sensitivity of the aerosol forcing to moderate changes of aerosol size has been discussed in detail by Pilinis et al. (1995).

In a third sensitivity test we assumed that BC was always externally mixed with the other components in each size range, forming pure BC spheres. 73% of the PMCAMx monthly mean AOD values changed by less than 0.01 in this test. The average change of the monthly mean AOD was negligible (< 0.5%). These two cases represent relative extreme cases of mixing. The results are reasonable given the relatively low levels of BC and the high levels of secondary inorganic and organic aerosol during this period. A similarly low sensitivity of the order of 1% or so is expected in the case of a core-shell model which was not examined in detail.

## 7 Conclusions

Previous evaluations of the ability of the 3-D CTM PMCAMx to reproduce the aerosol levels in Europe, the US, and Mexico City have been based on comprehensive chemical composition measurements at a few ground sites and limited data from a few flights. In this study we expand these efforts by using the MODIS and AERONET retrievals of AOD over Europe during a photochemically active period (May 2008). We exclude periods during which the different areas are strongly affected by dust (mainly from the Sahara) or other coarse particles like sea-salt in an effort to focus on the other primary and secondary anthropogenic and biogenic aerosol components.

PMCAMx can reproduce the observed AODs for this period with little bias (-16% for MODIS and +4% for AERONET). The corresponding fractional errors are 33% against MODIS and 22% against AERONET. These results are consistent with those of Fountoukis et al. (2011; 2014) who compared the PMCAMx predictions for the same period against ground measurements of fine PM composition in four sites and airborne measurements from several flights over central and northern Europe.

The AOD performance of PMCAMx against the MODIS retrievals is "excellent", based on the Morris et al. (2005) performance criteria discussed above, in the Iberian Peninsula, UK/Ireland, central Europe, Russia-Belarus-Ukraine, Turkey-northern Africa. We were expecting significant discrepancies between predicted and observed AOD over Russia given the uncertainty in the corresponding emissions. However, the agreement was quite good with both AERONET and MODIS. This rather

surprising result clearly requires additional investigation and could be due to offsetting errors. The model performance is "good" based on the same criteria in East Europe, the Balkans, and over the Mediterranean, the North Atlantic, and the Black Sea. Finally, its performance is "average" in the relatively clean area of North Europe and the South Atlantic. The performance is more or less similar against AERONET with the exception of a few areas with only one or two AERONET stations. The average performance against the AERONET measurements is considered using the above criteria "excellent" and against MODIS it is on the borderline between "good" and "excellent".

The above results suggest that the major weaknesses of PMCAMx appear to be overpredictions of sulfate and/or organics over North and East Europe, underprediction of sulfate over the Balkans, and underprediction of fine sodium chloride, sulfates, or organics in the southern Mediterranean and South Atlantic. However, these discrepancies are quite sensitive to the relative humidity fields predicted by WRF. In a sensitivity test the average predicted AOD increased by 13% (ranging from 7 to 31% depending on the area) for a uniform 5% change in RH. On the other hand, the details of the fine PM size distribution and the black carbon mixing state have a very small effect on the AOD predictions.

Comparison of the predicted AOD with the MODIS and AERONET results can shed only limited light on the ability of a CTM to reproduce the composition of the aerosol. The performance of the model for AOD, combined with its performance for composition in the sites where there are ground and airborne PM composition measurements, can be used to strengthen these tentative conclusions about its composition performance. These are clearly limited to the components dominating the AOD in each area, as discussed above, and either suggest problems or lack of major errors.

**Code availability**

PMCAMx is the research version of the publicly available CAMx (www.camx.org). The Fortran source code of CAMx (Version 6.20 was posted on March 23, 2015) and a User's Guide both prepared by ENVIRON can be downloaded through the above website. The PMCAMx code is used as testbed for testing of different hypotheses, algorithms, etc. The version used in this paper as well as the most current version can be obtained upon request by contacting Prof. S. Pandis (spyros@chemeng.upatras.gr).

*Acknowledgements.* The NASA MODIS team is acknowledged for preparing and making available MODIS observations. The AERONET team is acknowledged for establishing and maintaining the AERONET sites used in this study. Funding was provided by the FP7 ERC IDEAS project ATMOPACS.

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

| Region | Number of AERONET stations | Mean AERONET AOD | Mean PMCAMx AOD | Mean Error | Mean Bias | Fractional Error | Fractional Bias |
|---|---|---|---|---|---|---|---|
| UK/Ireland | 1 | 0.25 | 0.24 | 0.01 | -0.01 | 0.04 | -0.04 |
| Central Europe | 25 | 0.16 | 0.17 | 0.03 | 0.02 | 0.22 | 0.12 |
| North Europe | 4 | 0.08 | 0.12 | 0.04 | 0.04 | 0.36 | 0.36 |
| Spain and Portugal | 8 | 0.11 | 0.12 | 0.02 | 0.01 | 0.20 | 0.05 |
| East Europe | 2 | 0.11 | 0.14 | 0.03 | 0.03 | 0.24 | 0.24 |
| Balkans | 2 | 0.21 | 0.15 | 0.06 | -0.06 | 0.33 | -0.33 |
| Russia, Belarus, and Ukraine | 5 | 0.15 | 0.14 | 0.02 | -0.02 | 0.16 | -0.11 |
| Turkey and Northern Africa | 2 | 0.17 | 0.12 | 0.05 | -0.05 | 0.30 | -0.30 |
| Mediterranean Sea | - | - | - | - | - | - | - |
| North Atlantic Ocean | 1 | 0.11 | 0.16 | 0.05 | 0.05 | 0.37 | 0.37 |
| South Atlantic Ocean | - | - | - | - | - | - | - |
| Black Sea | - | - | - | - | - | - | - |
| Domain | 50 | 0.15 | 0.15 | 0.03 | 0.001 | 0.22 | 0.04 |

$$\text{Mean Error} = \frac{1}{N}\sum_{i=1}^{N}\left|P_i - O_i\right| \qquad \text{Mean Bias} = \frac{1}{N}\sum_{i=1}^{N}\left(P_i - O_i\right)$$
$$\text{Fractional Error} = \frac{2}{N}\sum_{i=1}^{N}\frac{\left|P_i - O_i\right|}{P_i + O_i} \qquad \text{Fractional bias} = \frac{2}{N}\sum_{i=1}^{N}\left(\frac{P_i - O_i}{P_i + O_i}\right)$$
where $P_i$ are predicted values by PMCAMx, $O_i$ the AERONET retrievals and $N$ the number of stations.

**Table 2.** Error metrics for the evaluation of PMCAMx against MODIS monthly mean AODs.

| Region | Mean MODIS AOD | Mean PMCAMx AOD | Mean Error | Mean Bias | Fractional Error | Fractional Bias |
|---|---|---|---|---|---|---|
| UK and Ireland | 0.21 | 0.23 | 0.04 | 0.02 | 0.22 | 0.14 |
| Central Europe | 0.16 | 0.17 | 0.05 | 0.01 | 0.30 | 0.13 |
| North Europe | 0.09 | 0.14 | 0.06 | 0.05 | 0.53 | 0.47 |
| Spain and Portugal | 0.14 | 0.12 | 0.04 | -0.03 | 0.28 | -0.15 |
| East Europe | 0.13 | 0.15 | 0.05 | 0.03 | 0.35 | 0.25 |
| Balkans | 0.19 | 0.14 | 0.05 | -0.04 | 0.28 | -0.24 |
| Russia, Belarus, and Ukraine | 0.13 | 0.12 | 0.04 | 0.01 | 0.30 | -0.01 |
| Turkey and Northern Africa | 0.16 | 0.14 | 0.05 | -0.03 | 0.31 | -0.13 |
| Mediterranean Sea | 0.18 | 0.14 | 0.04 | -0.04 | 0.25 | -0.24 |
| North Atlantic Ocean | 0.17 | 0.14 | 0.04 | -0.03 | 0.30 | -0.21 |
| South Atlantic Ocean | 0.16 | 0.10 | 0.06 | -0.06 | 0.45 | -0.45 |
| Black Sea | 0.17 | 0.14 | 0.04 | -0.03 | 0.22 | -0.18 |
| Domain | 0.16 | 0.14 | 0.05 | -0.02 | 0.33 | -0.16 |

**Table 3.** Monthly predicted mean ground-level concentration in μg m$^{-3}$ of the major PM$_{2.5}$ components

| Region | SO$_4^{2-}$ | OA | EC | Cl$^-$ | Na$^+$ | NH$_4^+$ | NO$_3^-$ | H$_2$O | Crustal |
|---|---|---|---|---|---|---|---|---|---|
| UK and Ireland | 3.6 | 3.4 | 0.5 | 0.6 | 0.8 | 2.4 | 3.8 | 33.3 | 0.7 |
| Central Europe | 3.0 | 3.4 | 0.5 | 0.2 | 0.4 | 1.4 | 1.3 | 10.1 | 0.6 |
| North Europe | 2.2 | 2.3 | 0.2 | 0.2 | 0.4 | 0.9 | 0.6 | 5.1 | 0.4 |
| Spain and Portugal | 1.6 | 1.2 | 0.2 | 0.1 | 0.2 | 0.7 | 0.5 | 11.7 | 0.3 |
| East Europe | 2.9 | 3.1 | 0.4 | 0.1 | 0.4 | 1.2 | 0.8 | 9.2 | 0.6 |
| Balkans | 3.9 | 2.7 | 0.3 | 0.1 | 0.3 | 1.4 | 0.3 | 4.9 | 0.6 |
| Russia, Belarus, and Ukraine | 2.5 | 2.1 | 0.3 | 0.03 | 0.2 | 0.9 | 0.2 | 3.9 | 0.5 |
| Turkey and Nothern Africa | 2.7 | 2.2 | 0.2 | 0.2 | 0.4 | 1.0 | 0.6 | 8.2 | 0.5 |
| Mediterranean Sea | 4.2 | 2.4 | 0.3 | 1.3 | 1.4 | 1.2 | 0.3 | 11.3 | 0.7 |
| North Atlantic Ocean | 2.1 | 1.8 | 0.2 | 1.0 | 1.0 | 1.0 | 1.0 | 24.2 | 0.4 |
| South Atlantic Ocean | 1.5 | 1.1 | 0.06 | 0.8 | 0.8 | 0.5 | 0.3 | 8.8 | 0.3 |
| Black Sea | 3.8 | 2.8 | 0.3 | 0.6 | 0.7 | 1.3 | 0.4 | 9.7 | 0.7 |
| Domain | 2.4 | 1.9 | 0.2 | 0.5 | 0.6 | 0.9 | 0.5 | 10 | 0.5 |

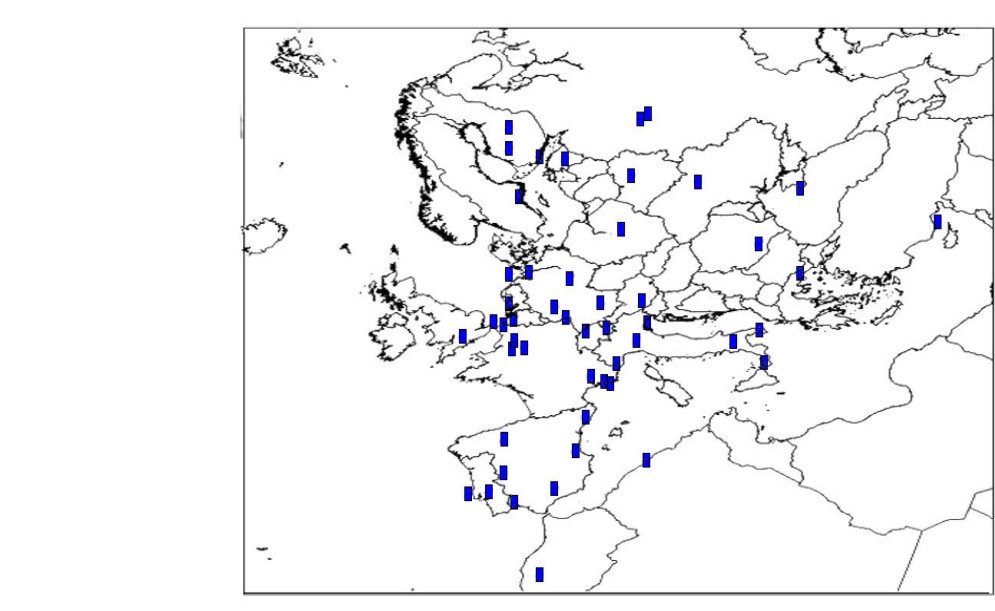

**Figure 1.** Geographical distribution of the 50 AERONET stations used in the present study.

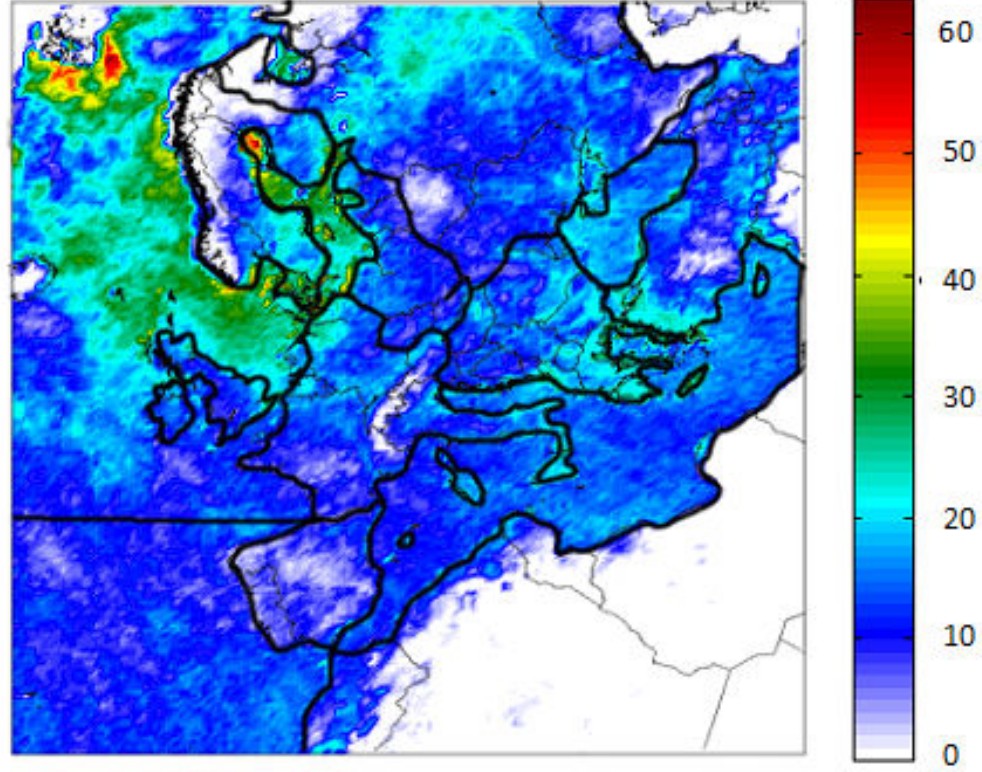

**Figure 2.** Geographical distribution of the number of available AOD retrievals from MODIS over Europe during May 2008. White color denotes no retrievals. Land is partitioned into 8 regions including the United Kingdom and Ireland, central Europe, North Europe, Spain and Portugal, East Europe, Balkans, Russia/Belarus/Ukraine, Turkey, and Northern Africa. The sea is partitioned into 4 regions: the Mediterranean, North Atlantic, South Atlantic, and Black Sea.

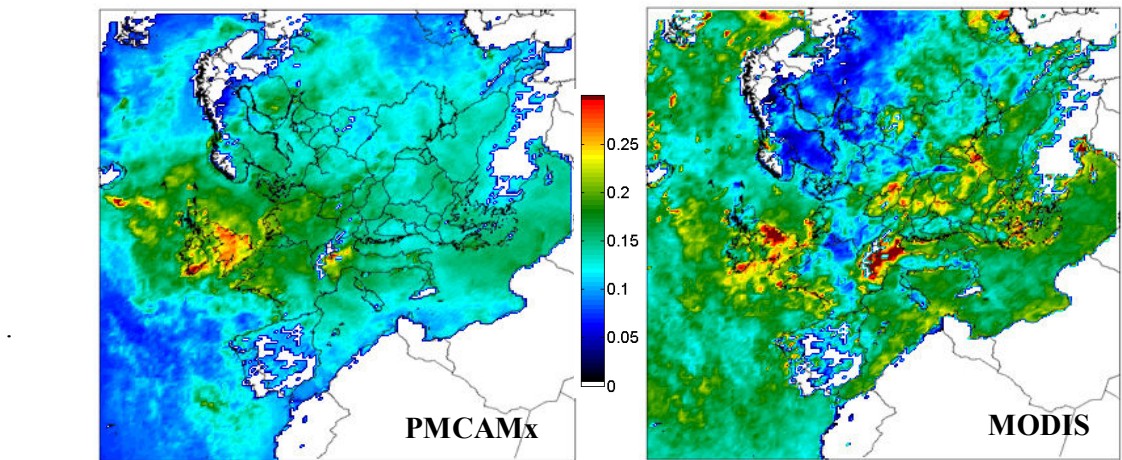

**Figure 3.** Monthly mean AODs from PMCAMx and MODIS (QAC ≥ 1) during May 2008. White color denotes no AOD retrieval. A coarse particle rejection filter has been employed. The PMCAMx AODs correspond to the periods of the MODIS retrievals.

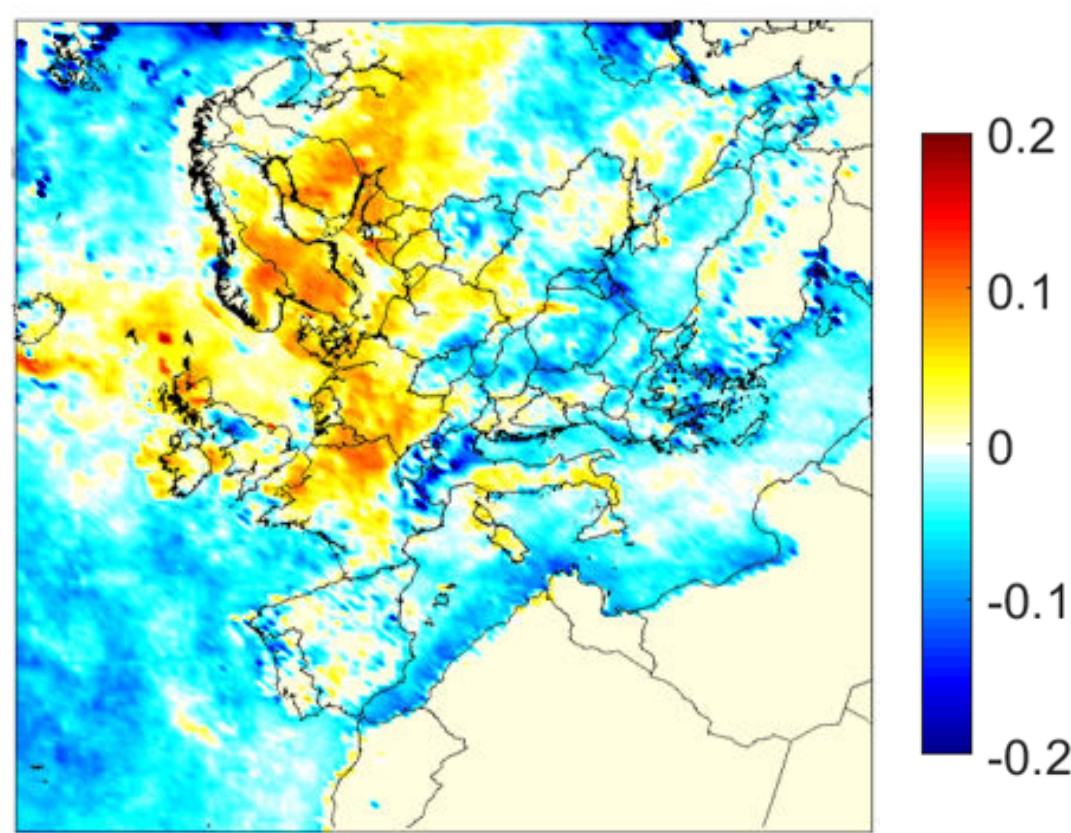

**Figure 4.** Difference of the PMCAMx from MODIS (QAC ≥ 1) monthly mean AODs during May 2008. Positive means that PMCAMx overpredicts AOD compared to MODIS. There were not enough dust-screened AOD retrievals for the model evaluation in the white areas in Northern Africa and in the Middle East.

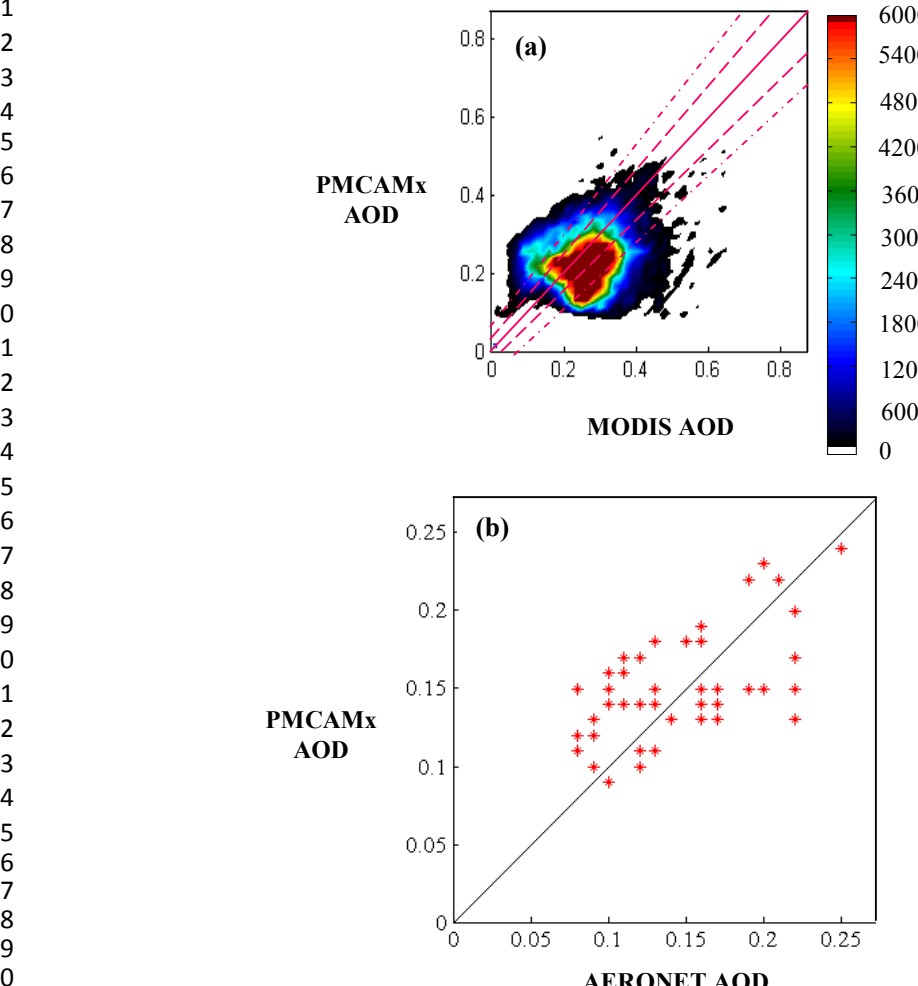

**Figure 5. a)** Comparison of the PMCAMx predictions with MODIS (QAC ≥ 1) monthly mean AODs. The different colors indicate density. The dashed red lines denote the ocean expected error envelope and the dotted lines denote the land envelope which describes MODIS AOD uncertainties with respect to AERONET (see Section 3). The solid red line is the 1:1 line. **b)** Comparison of the PMCAMx with AERONET monthly mean AODs. The PMCAMx values correspond to the periods of measurement for the 50 AERONET stations.

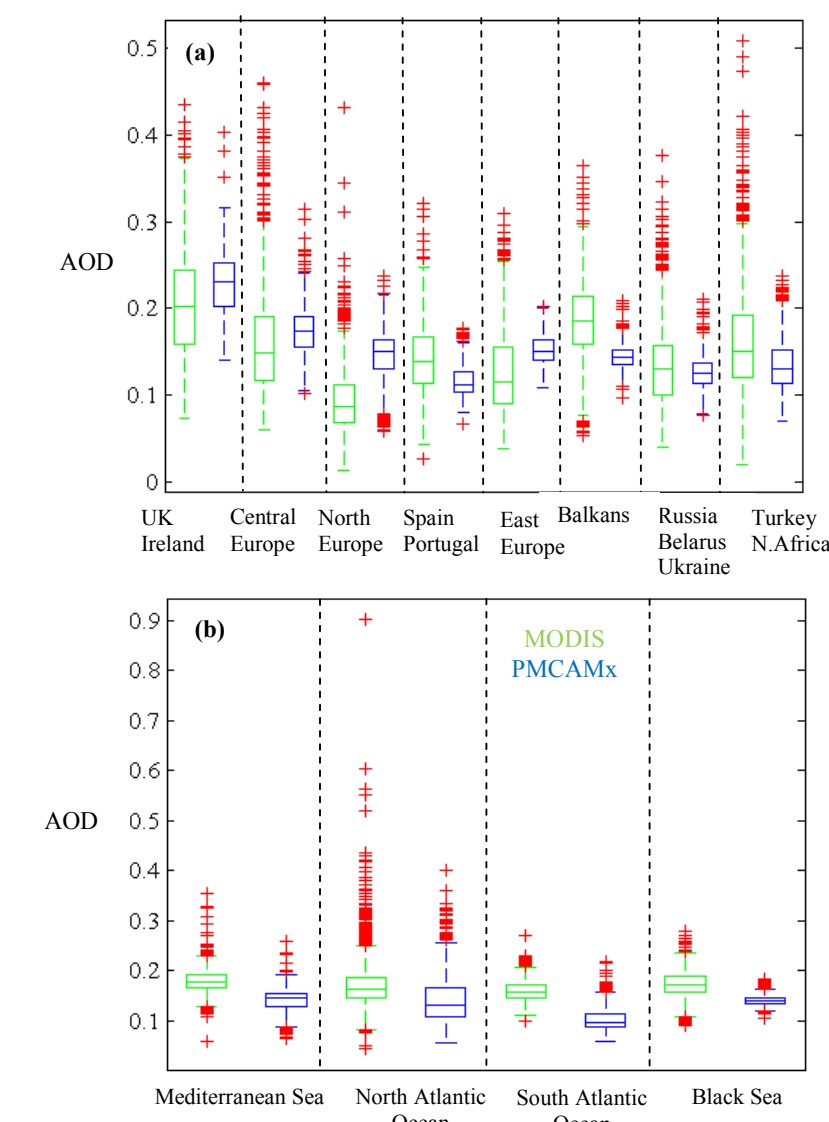

**Figure 6. a)** Box plots of the PMCAMx and MODIS (QAC ≥ 1) monthly mean AODs for land. The central mark is the median, the edges of the box are the 25th and 75th percentiles, the whiskers extend to the extreme data points considered to be not outliers, and the outliers are plotted individually by the red marks. Points are drawn as outliers if they are larger than Q3+1.5(Q3-Q1) or smaller than Q1-1.5(Q3-Q1), where Q1 and Q3 are the first and third quartiles, respectively. **b)** Box plots of the PMCAMx and MODIS (QAC ≥ 1) monthly mean AODs for water.