# Peer review of "Comparison of PMCAMx aerosol optical depth predictions over Europe with AERONET and MODIS measurements"

_Geoscientific Model Development, 2015_

## Referee Comment (RC1) · Anonymous Referee #1 · 25 Feb 2016

The manuscript evaluates the PMCAMx model, comparing the aerosol optical depth (AOD) simulated by the model with observations from MODIS and AERONET. The manuscript fits perfectly the goals of the GMDD journal and the methodology and results are clear.

However, as there are no space limitation for this journal, and being GMD(D) dedicated to technical and specific publication I would have expected to have a detail and complete description of the modeling system and the observational datasets used for the evaluation. However, to my point of view, this was not the case.

Although I have no real comments on the methodology and results, the lack of detailed informations raises serious doubts on the scientific relevance of such an evaluation.

The authors should therefore add all the necessary informations to the manuscript before this can be considered for publication.

**Major comments:**

**Aerosol concentration evaluation:** The aerosol optical depth can be considered as indirect method to evaluate the model performance, as this is normally estimated from the aerosol composition and the radiative properties of the aerosol components. Therefore I find quite disturbing the absolute lack of any discussion in the capability of the model to reproduce the observed aerosol composition and concentration before to evaluate the AODs. Wrong aerosols compositions could still give reasonable AODs, but for the wrong reasons. Therefore I urge the authors to evaluate also their aerosol composition results. For example, in the introduction the authors stated that "these errors are probably due to an underestimation of sulfates". I expect to be enough sulfate measurements in Europe to validate this statement, as example with the AirBase dataset (http://acm.eionet.europa.eu/databases/airbase/) which present up to hourly observations for single stations. Additionally, only PM1 evaluation of PMCAMx is mentioned, although this was "limited in space" (page 4, line 15). Therefore I would recommend to first have a through evaluation of the aerosols fields against measurements (AirBase, EMEP...) before to dig into the detail of the AOD. If this was probably published elsewhere, it is impossible to find such reference in this manuscript, also in the PCMCAMx description. In the conclusions it has been mentioned that PMCAMx aerosols composition simulation was evaluated, but no publications have been listed.

**Period of simulation/analysis:** This analysis is focusing on the period 1-29 of May 2008. The first information on the period is on page 8, line 15, under the satellite description, which is possibly not the best location for such information. Nevertheless, it is somehow unclear to me why this period has been chosen. Why not the

entire month of May? Why not another period of the 2008?

Linked to this issue is also the poor description on how PMCAMx has been used to simulate the period of interest. As I am not familiar with the modeling system, is it difficult to me to understand the first sentence of page 6 "To limit the effect of the initial conditions on the results, the first six days of *each* simulation were excluded from the analysis". Are you referring to multiple simulations? Is the model re-initialized? Or was it a continuous simulation from which only May 2008 was extracted? These informations are essential to put the model into context, but they are largely missing in the manuscript. Possibly few references would help the reader to gather the missing informations, if the description of the model set-up would be too tedious. Nevertheless these are simply not there.

Finally, it would have been interesting to make an analysis of an entire year, so to cover the different dynamical and chemical space, such as strong aerosol emissions in winter and strong photochemistry in summer. If that is a difficult task, at least few time-slice analysis for different seasons should be performed.

**MODIS data:** The author are using MODIS data collection 5.1. Although newer products are available since early 2015 (collection 6), it would be good to know exactly which products you are using. If I am not wrong, in the MODIS collection 5.1 in the AQUA platform both Deep-Blue and Dark-Target algorithm are available. Which one did you used? Additionally, you used "the union of Terra and Aqua MODIS AOD [..]" Could you explain what do you mean with union? How did you unified the two fields? Finally, you are using level 2 data and you calculated the monthly average for May 2008. Which spatial resolution did you used to create such field? How did you merge spatially the observations? As you were using monthly averages, why not using using level 3 data? There is a severe lack of informations here that are important to understand how these sensed AODs have been produced. I strongly suggest to fully rewrite the section with the additional informations.

**Minor comments:**

**Title:** Why PMCAMx-2015? Is that a new version of PMCAMx? If so, it would be great to use the same naming convention through all the manuscript.

**Page 5, line 16:** The PMCAMx model is using results from WRF as meteorological forcing. Which frequency is needed? Could the author add some comments on the possible error introduced by the non exact dynamics? Is there any evaluation of the dynamics?

**Page 5, line 20:** Same for the emissions: is there any reference and comparison with other emissions dataset?

**Page 6, line 1:** Would be good to mention here the period covered by the simulation(s). Here reads as there are more than one. Could the author be more specific? (see major comments).

**Page 9, line 22** What do you mean with "The PMCAMx AODs have been calculated fir exactly the same period as MODIS retrivals[...]?" Are you using model results at TERRA/ACQUA overpass (local time)? Are you using daily average for the periods where observations are available? Please specify.

**Page 10, line 2** Does it make sense to compare this region when most of the data are masked due to the strong presence of dust aerosols there? Your data sample is strongly reduced, probably not allowing a great statistics here. The same is valid for Turkey and North Africa region.

**Fig.4** You mentioned that the white areas means that not enough dust-screened AODs are present. However, it seems to me that in the Po valley the white area is much larger that what is present in Fig 3 from MODIS and PMCAMx. Are you sure that here you are not masking additional values?
**Remarks:**

To my knowledge the author "Meij" should read "de Meij". Please check the references.

---

## Referee Comment (RC2) · Anonymous Referee #2 · 27 Feb 2016

Review on manuscript

PMCAMx-2015 evaluation over Europe against AERONET and MODIS aerosol optical depth measurements by A. Panagiotopoulou, P. Charalambidis, C. Fountoukis, C. Pilinis, and S. N. Pandis

Remote sensing measurements of aerosols represent a valuable complementary to surface in-situ data for CTM evaluation. Indeed, satellite observations provide finely resolved in space AOD data with global coverage, though being of somewhat varying quality due to assumptions involved in the retrieval algorithms. AERONET sun-photometers provide directly measured AOD at high time-resolution. Therefore last decades those data have been increasingly widely used for model evaluations.

[Figure]

In this work, the authors make use of MODIS and AERONET measured AOD to compare with results from PMCAMx-2015 model in order to get better insight in the model performance with respect to aerosol loads. Thus, the paper addresses relevant to the scope of GMD issues.

The article is very neatly and clearly written, and the methods applied are valid, but it does not offer any substantial novelty regarding ideas, data or methodology.

Some of the conclusions appear not to be satisfactorily well founded (i.e. regarding model performance with respect to the individual aerosol types based on AOD evaluation).

The title contain a proper reference to the model used, but does not indicate the short-term (one month) and thus limited model evaluation. Besides, only levels of monthly mean AOD have been compared, rather than a complete evaluation. Therefore, I'd suggest to use "comparison" instead of "evaluation". Also, I'd not advise to include rather hypothetical explanations (lines 22-25), but rather say that the probable reasons of disagreements are discussed in the paper.

In general, the paper is written in good language, the formulations are clear and the supplemented references are relevant and ample.

Other comments:

1. The considered period (May 2008) should be indicated in the Abstract and in Sections 2, 3.

2. I recommend to include a bit more complete summary of earlier evaluation of all aerosol components

3. Explain more clearly whether the model calculates size-resolved chemical composition, or only size resolved number density

4. For comprehensive and robust model evaluation and better understanding model

result more in-depth analysis should be performed, including spatial and temporal correlations, RMSE, STD etc etc.

4. I find the explanations of model vs observations AOD discrepancies by over/underestimation of a particular aerosol components a bit speculative. i would strongly recommend to also include (at least) aerosol evaluation with monitoring surface data in different regions (and airborne measurements if possible) to supposr the conclusions.

P. 2 lines 13-14: What is the temporal resolution of AERONET data?

P. 4 lines 13-16: provide biases for all aerosol species and even better for the regions included in your AOD discussion; only 4 sites with data for sulphates???

P. 7 line 3: How is Mie theory applied for aerosol mass? line 10: Have you made tests on accounting for "brown carbon", i.e. absorbing OC (which is believed to make notable contribution)? Lines 19... Study period? time resolution of AERONET data? AOD at which wave length was used?

P. 8 line 7: location instead of part

P. 9 lines 4-6: I do not understand. Suggest the explain better, or just refer to the sources. Lines 22-23: times coinciding with the satellites' overpasses?

P. 10 line16: compared with

pp. 11 lines 10-18: Given rather poor quality of emission data for those regions, I feel rather skeptic and "alarmed" about good agreement between model and measurements

p. 13 line 4: Rather sloppy formulation

P. 15 line 16-18: This is rather unfair statement. MODIS data is particularly valuable due to its spatial coverage (besides the AOD errors are relatively small) line 16: correct "complement" line 21-22: Please, elaborate , otherwise leave out. It's not needed

unless model comparison with MODEl and AEROCOM lead to different conclusions.

P 16 line 7: again "excellent" model performance using poor emission input is typically indicative of some kind of compensating errors. Lines 15-17: too speculative conclusion about model's excelling in calculating all of aerosol types

―――――――――――――――

---

## Author Comment (AC1) · 3 Apr 2016

**(1)** *The manuscript evaluates the PMCAMx model, comparing the aerosol optical depth (AOD) simulated by the model with observations from MODIS and AERONET. The manuscript fits perfectly the goals of the GMDD journal and the methodology and results are clear. However, as there are no space limitations for this journal, and being GMD(D) dedicated to technical and specific publication I would have expected to have a detail and complete description of the modeling system and the observational datasets used for the evaluation. However, to my point of view, this was not the case. Although I have no real comments on the methodology and results, the lack of detailed information raises serious doubts on the scientific relevance of such an evaluation. The*

*authors should therefore add all the necessary information to the manuscript before this can be considered for publication.*

We appreciate the constructive comments of the referee. We have followed the corresponding suggestions adding more information about the model and its inputs. There have been more than 10 papers that have been published describing PMCAMx and its evolution, so we necessarily rely on the corresponding references for a lot of the details. We believe that adding a complete and detailed model description (something that would require probably hundreds of pages) to every GMD paper would be clearly problematic.

*Major Comments:*

**(2)** *Aerosol concentration evaluation: The aerosol optical depth can be considered as indirect method to evaluate the model performance, as this is normally estimated from the aerosol composition and the radiative properties of the aerosol components. Therefore, I find quite disturbing the absolute lack of any discussion in the capability of the model to reproduce the observed aerosol composition and concentration before to evaluate the AODs. Wrong aerosols compositions would still give reasonable AODs, but for the wrong reasons. Therefore, I urge the authors to evaluate also their aerosol composition results. For example, in the introduction the authors stated that "these errors are probably due to an underestimation of sulfates". I expect to be enough sulfate measurements in Europe to validate this statement, as example with the AirBase dataset (http://acm.eionet.europa.eu/databases/airbase/) which present up to hourly observations for single stations. Additionally, only PM1 evaluation of PMCAMx is mentioned, although this was "limited in space" (page 4, line 15). Therefore, I would recommend to first have a through evaluation of the aerosols fields against measurements (AirBase, EMEP..) before to dig in the detail of AOD. If this was probably published elsewhere, it is impossible to find such reference in this manuscript, also in the PMCAMx description. In the conclusions it has been mentioned that PMCAMx aerosols*

[Figure]

*composition simulation was evaluated, but no publications have been listed.*

The reviewer has unfortunately missed the following statement in lines 11-14 of page 4 of the original manuscript: "The $PM_1$ composition predictions of PMCAMx have been evaluated over Europe in May 2008 (Fountoukis et al., 2011). PMCAMx performance against airborne measurements was as good as its performance against the hourly ground measurements. More than 94 percent of the organic aerosol (OA) hourly values and more than 82 percent of the sulfate ones were reproduced within a factor of 2." This paper provides a detailed evaluation of the ability of PMCAMx to reproduce the detailed EUCAARI campaign PM composition measurements over Europe. This includes both ground measurements and airborne measurements during the EUCAARI flights. There were approximately 8500 measurements (data points) that were used in this evaluation. Please note that this is the same period as the one analyzed in the present work. We do not believe that the referee's statement about "the absolute lack of any discussion in the capability of the model to reproduce the observed aerosol composition and concentration" is justified.

In this work we exclude the periods with high dust (or in general coarse particles) concentrations, so $PM_1$ is the appropriate metric for composition evaluation. In the Fountoukis et al. (2011) paper we have used all the available $PM_1$ composition measurements in Europe for the corresponding period. This is better clarified in the revised paper.

A more detailed evaluation of the ability of PMCAMx to reproduce the organic aerosol composition during the same period has been published by Fountoukis et al. (Organic aerosol concentration and composition over Europe: insights from comparison of regional model predictions with aerosol mass spectrometer factor analysis, ACP, 9061-9076, 2014). A discussion of the findings of this evaluation exercise has been added to the revised paper.

We have added a new section in the paper focusing on just the published evaluations

of the capability of PMCAMx to reproduce the PM composition and concentration over Europe to make sure that similar misunderstandings can be avoided.

**(3)** *Period of simulation/analysis: The analysis is focusing on the period 1-29 May 2008. The first information on the period is on page 8, line 15, under the satellite description, which is possibly not the best location for such information. Nevertheless, it is somehow unclear to me why this period has been chosen. Why not the entire month of May? Why not another period of the 2008? Linked to this issue is also the poor description on how PMCAMx has been used to simulate the period of interest. As I am not familiar with the modeling system, it is difficult to me to understand the first sentence of page 6 "To limit the effect of the initial conditions on the results, the first six days of each simulation were excluded from the analysis". Are you referring to multiple simulations? Is the model re-initialized? Or was it a continuous simulation from which only May 2008 was extracted? These pieces of information are essential to put the model into context, but they are largely missing in the manuscript. Possibly few references would help the reader to gather the missing information, if the description of the model set-up would be too tedious. Nevertheless there are simply not there. Finally, it would have been interesting to make an analysis of an entire year, so to cover the different dynamical and chemical space, such as strong aerosol emissions in winter and strong photochemistry in summer. If that is a difficult task, at least few time-slice analysis for different seasons should be performed.*

The May 2008 period was chosen for two reasons. First this was the period of the EU-CAARI campaign focusing on a photochemically active period with summertime like concentrations. Detailed continuous measurements of $PM_1$ composition both at the ground and aloft as well as a corresponding emission inventory (prepared by TNO) exist for that period. The second reason was that the ability of PMCAMx to reproduce these detailed $PM_1$ composition measurements has already been evaluated in previous work (Fountoukis et al., 2011; 2014) and therefore we could focus on the optical properties of the fine particulate matter in this paper. The exact dates simulated were

the same as in the previous publications for consistency. We have tried to clarify the reasons for this selection in the revised paper.

We have clarified the initialization procedure of the model. Given that the initial conditions are quite uncertain and the first few days are dominated by them, we are excluding the corresponding "start-up" period from the model evaluation.

Only one baseline model simulation was performed together with a number of sensitivity tests described in the paper. The corresponding confusing sentence has been rephrased.

We do intend to extend the current work to other seasons. However, each season is characterized by its own issues. For example, during winter there is significant wood burning activity over Europe and the corresponding emissions are probably seriously underestimated in a lot of countries. We have tried to address this issue recently in Denier van der Gon et al. (ACP, 15, 6503-6519) where we try to improve the wood burning emission inventories for the wintertime. The current work focuses on a photochemically active period with the AOD dominated by mostly secondary fine aerosol.

**(4)** *MODIS data: The authors are using MODIS data collection 5.1. Although newer products are available since early 2015 (collection 6), it would be good to know exactly which products you are using. If I am not wrong, in the MODIS collection 5.1 in the AQUA platform both Deep-Blue and Dark-Target algorithm are available. Which one did you use? Additionally, you used "the union of Terra and Aqua MODIS AOD [...]". Could you explain what do you mean with union? How did you unify the two fields? Finally, you are using level 2 data and you calculated the monthly average for May 2008. Which spatial resolution did you use to create such field? How did you merge spatially the observations? As you were using monthly averages, why not using level 3 data? There is a severe lack of pieces of information here that are important to understand how these sensed AODs have been produced. I strongly suggest to fully rewrite the section with the additional information.*

We have followed the reviewer's suggestion and we have rewritten the corresponding section of the paper, clarifying the MODIS products used and their processing for the evaluation in an effort to avoid any ambiguities.

Briefly, regarding the details, the Dark-Target algorithm products were used. By "union" we mean that data from both Terra and Aqua datasets were used in order to have better spatial and temporal coverage. We did not alter the values of the data records and we did not apply any sort of transformations. We have changed the corresponding text to make these clear to the reader. The MODIS AOD values, retrieved with spatial resolution 10x10km$^2$, were collocated onto the grid of the PMCAMx modeling domain. Over the May 2008 simulation here were several values of MODIS AOD attributed to each cell of PMCAMx. Then, the monthly mean AOD for each grid cell was calculated by taking the average value of the MODIS AODs falling inside it. The L2 MODIS data are better suited to the temporal and spatial resolution of our model. Hourly PMCAMx concentration predictions were used are. Also, the filtering of the MODIS retrievals over Land and Ocean is performed prior to monthly averaging. Therefore, it is necessary to use L2 MODIS data since they have the necessary temporal resolution. In any case L3 MODIS data are derived from the L2 data.

Minor Comments:

**(5)** *Title: Why PMCAMx-2015? Is that a new version of PMCAMx? If so, it would be great to use the same naming convention through all the manuscript.*

We have replaced "PMCAMx-2015" with "PMCAMx" in the title and throughout the manuscript to avoid unnecessary confusion.

**(6)** *Page 5, line 16: The PMCAMx model is using results from WRF as meteorological forcing. Which frequency is needed? Could the author add some comments on the possible error introduced by the non exact dynamics? Is there any evaluation of the dynamics?*

We now explain that we used hourly meteorological data from WRF as input to PM-CAMx. We have also added details about the application of WRF. The performance of WRF for several air quality-relevant parameters (wind speed, wind direction, RH, temperature) in the same domain is discussed by Fountoukis et al. ACPD, 2016). The agreement with measurements was more than satisfactory.

**(7)** *Page 5, line 20: Same for the emissions: is there any reference and comparison with other emissions dataset?*

References for the development of the EUCAARI TNO emissions have been added.

**(8)** *Page 6, line 1: Would be good to mention here the period covered by the simulation(s). Here reads as there are more than one. Could the author be more specific? (see major comments).*

The first sentence in page 6 has been corrected to "To limit the effect of the initial conditions on the results, the first six days of the basecase simulation were excluded from the analysis". Only one baseline model simulation for May 2008 was performed. We have included this information in the text to make it clear to the reader.

**(9)** *Page 9, line 22: What do you mean with "The PMCAMx AODs have been calculated for exactly the same period as MODIS retrievals[...]?" Are you using model results at TERRA/ACQUA overpass (local time)? Are you using daily average for the periods where observations are available? Please specify.*

To evaluate PMCAMx performance for the period of May 2008 we only used model results that correspond to the same Terra/Aqua overpass time. Monthly averages are calculated from the corresponding PMCAMx AODs and the MODIS retrievals. We do not use daily average values to make the comparison more exact. We have made the proper changes to the text in order to clarify this important point.

**(10)** *Page 10, line 2: Does it make sense to compare this region when most of the data are masked due to the strong presence of dust aerosols there? Your data sample is strongly reduced, probably not allowing a great statistics here. The same is valid for Turkey and North Africa region.*

This is a valid concern by the reviewer. We have added this point in the revised text. Since the data sample size is small in Turkey and North Africa the corresponding comparisons provide little information. This is the reason that we have avoided discussing these regions in any detail in the paper.

**(11)** *Fig. 4: You mentioned that the white areas mean that not enough dust-screened AODs are present. However, it seems to me that in the Po Valley the white area is much larger that what is present in Fig. 3 form MODIS and PMCAMx. Are you sure that here you are not masking additional values?*

This was due to our choice of colors and scale. We have redrawn Figure 4 correcting this problem pointed out by the referee.

*Remarks:*

**(12)** *To my knowledge the author "Meij" should read "de Meij". Please check the references.*

We thank the reviewer for the correction. We have replaced "Meij" with "de Meij".

---

## Author Comment (AC2) · 3 Apr 2016

**(1)** *Remote sensing measurements of aerosols represent a valuable complementary to surface in-situ data for CTM evaluation. Indeed, satellite observations provide finely resolved in space AOD data with global coverage, though being of somewhat varying quality due to assumptions involved in the retrieval algorithms. AERONET sun-photometers provide directly measured AOD at high time-resolution. Therefore, last decades those data have been increasingly widely used for model evaluations. In this work, the authors make use of MODIS and AERONET measured AOD to compare with results from PMCAMx-2015 model in order to get better insight in the model performance with respect to aerosol loads. Thus, the paper addresses relevant to the*

*scope of GMD issues.*

*The article is very neatly and clearly written, and the methods applied are valid, but it does not offer any substantial novelty regarding ideas, data or methodology. Some of the conclusions appear not to be satisfactorily well founded (i.e. regarding model performance with respect to the individual aerosol types based on AOD evaluation).*

*The title contain a proper reference to the model used, but does not indicate the short term (one month) and thus limited model evaluation. Besides, only levels of monthly mean AOD have been compared, rather than a complete evaluation. Therefore, I 'd suggest to use "comparison" instead of "evaluation". Also, I'd not advise to include rather hypothetical explanations (lines 22-25), but rather say that the probable reasons of disagreements are discussed in the paper.*

*In general, the paper is written in good language, the formulations are clear and the supplemented references are relevant and ample.*

We do appreciate all the comments and suggestions of the referee. The major new methodological improvement in this work is the screening of the satellite retrievals for periods with high dust (or coarse particles in general) concentrations and the combination of the MODIS/AERONET datasets so that the conclusions can be more robust. This is now stressed in the revised manuscript.

We have followed the reviewer's suggestion and changed the word "evaluation" with "comparison" in the title of the paper.

It is clear that comparison of the predicted AOD with the MODIS/AERONET results can shed only limited light on the ability of a CTM to reproduce the composition of the aerosol. We have rephrased the corresponding sentences in the conclusions stressing that the performance of the model for AOD (combined with its performance for composition in the sites where there are ground and airborne PM composition measurements) can be used to derive some tentative conclusions about its composition performance.

These are clearly limited to the components dominating the AOD in each area and either suggest problems or lack of major errors.

*Other Comments:*

**(2)** *The considered period (May 2008) should be indicated in the Abstract and in Sections 2, 3.*

We have added the considered period of May 2008 (EUCAARI campaign) in the Abstract and in Sections 2 and 3.

**(3)** *I recommend to include a bit more complete summary of earlier evaluation of all aerosol components.*

We have followed the reviewer's suggestion and added a new section in which we provide a more extensive summary of the results of the earlier published evaluations of PMCAMx for the same period focusing on PM composition (see also Comment 2 of Referee 1).

**(4)** *Explain more clearly whether the model calculates size-resolved chemical composition or only size-resolved number density.*

We now explain in the revised section 2 that PMCAMx simulates the composition of each size section and therefore predicts the size-resolved PM composition using in this application 10 size bins. PMCAMx calculates the aerosol number from the corresponding mass distribution while its sister model, PMCAMx-UF, simulates both the aerosol number and mass distributions explicitly.

**(5)** *For comprehensive and robust model evaluation and better understanding model result more in-depth analysis should be performed, including spatial and temporal correlations, RMSE, STD etc.*

We have calculated additional performance metrics for the model including the RMSE and STD. These provide limited additional insights compared to the four metrics that are currently used in the paper. This information has been added to the Supplementary Material. We agree that the spatial dependence of the performance of the model is useful. We found that the separation of the model domain in areas, given our emphasis on secondary aerosol, was the best way to approach this issue. For the temporal performance we have added in the revised paper some discussion focusing mainly on the average diurnal profiles of the AERONET AOD.

**(6)** *I find the explanations of model vs observations AOD discrepancies by over/underestimation of a particular aerosol components a bit speculative. I would strongly recommend to also include (at least) aerosol evaluation with monitoring surface data in different regions (and airborne measurements if possible) to support the conclusions).*

We do agree that these explanations are necessarily speculative. The recommendation of the reviewer is very useful. We have combined the discussion of the AOD performance of the model with its composition performance for the areas (central Europe, United Kingdom and Ireland, North Atlantic, Mediterranean) in which there were PM composition measurements. Combining these date sources does strengthen our conclusions regarding the model performance in these areas.

**(7)** *P.2 lines 13-14: What is the temporal resolution of AERONET data?*

The AERONET measurements have a variable temporal resolution varying from 15 min when the sun is high up in the sky to higher values when the sun is closer to the horizon. Measurements start at sunrise when the sun is at approximately 7.5 degrees above the horizon and end at sundown when the sun is once more at approximately 7.5 degrees. This information has been added to Section 3 describing the AERONET data.

**(8)** *P.4 lines 13-16: provide biases for all aerosol species and even better for the regions included in your AOD discussion; only 4 sites with data for sulphates?*

We have added the biases for all aerosol species and analyzed them by region thus synthesizing the AOD and PM composition information. We have included the data from both the ground and the airborne measurements and therefore our comparison includes four regions and thousands of data points.

**(9)** *P.7 line 3: How is Mie theory applied for aerosol mass? line 10: Have you made tests on accounting for "brown carbon", i.e. absorbing OC (which is believed to make notable contribution)? Lines 19...Study period? time resolution of AERONET data? AOD at which wave length was used?*

We have added a paragraph and the corresponding references clarifying the application of Mie theory of the aerosol size composition distribution simulated by PMCAMx. We have tested in a sensitivity study the effect of the potential absorption enhancement of the BC due to coatings by the other PM components and the effect on AOD for this area and period has been found to be quite small. Given that the biomass burning emissions in Europe during that period were low and that biomass burning is expected to be one of the major sources of brown carbon the effect is also expected to be small. This is explained now in the revised paper. We also clarify in this page the study period (May 2008) and the AERONET AOD wavelength (550 nm). The variable AERONET data time resolution is discussed in our response to Comment 7 above.

**(10)** *P.8 line 7: location instead of part.*

We have replaced "part" with "location".

**(11)** *P.9 lines 4-6: I do not understand. Suggest to explain better, or just refer to the sources. Lines 22-23: times coinciding with the satellites' overpasses?*

We have rewritten this rather confusing sentence to explain better the binning of the data points for the comparison of the MODIS AOD with the AERONET AOD shown in Figure S1. The comparisons with the MODIS AOD retrievals correspond exactly in space and time, so the times coincide with the satellites' overpasses. We have made the corresponding clarification in the paper.

**(12)** *P.10 line 16: compared with.*

We have made the corresponding correction.

**(13)** *pp. 11 lines 10-18: Given rather poor quality of emission data for those regions, I feel rather skeptic and "alarmed" about good agreement between model and measurements.*

We were also expecting significant discrepancies between predicted and observed AOD over Russia given the uncertainty in the corresponding emissions. However, the agreement was quite good with both AERONET and MODIS. This rather surprising result clearly requires additional investigation and could be due to offsetting errors. This point is now stressed in the corresponding section.

**(14)** *p.13 line 4: Rather sloppy formulation.*

We have rewritten the corresponding sentence.

**(15)** *P.15 line 16-18: This is a rather unfair statement. MODIS data is particularly valuable due to its spatial coverage (besides the AOD errors are relatively small). Line 16: correct "complement" Line 21-22: please, elaborate, otherwise leave out. It's not needed unless model comparison with MODEl and AEROCOM lead to different conclusions.*

[Figure]

We agree with the reviewer about the value of the MODIS data and the enormous value of the spatial coverage of the corresponding dataset. This sentence has been rephrased. We have corrected the typo in Line 16 and have deleted the potentially confusing sentence in Lines 21-22.

**(16)** *P. 16 line 7: again "excellent" model performance using poor emission input is typically indicative of some kind of compensating errors. Lines 15-17: too speculative conclusion about model's excelling in calculating all of aerosol types.*

In the revised manuscript we repeat at this point the uncertainty of the emissions in this region and the potential existence of some form of compensating errors. We have rephrased the sentence in Lines 15-17 to avoid misinterpretation of the corresponding findings.

---

## Author Response (AR2)

**Responses to Reviewer #1**

**(1)** The paper reports the comparison of PMCAMx AOD predictions with AERONET and MODIS observations. The comparison is made for limited time (1 month) and presented only for monthly average values. Periods with high dust contributions have been screened out, so that the evaluation is expected to reflect mainly the model's ability to reproduce the organic and secondary inorganic aerosols. In addition to the PM concentration, AOD depends also on the PM composition, size distribution, mixing state and interaction with atmospheric humidity. Thus, there are many possible reasons for the model measurement discrepancies and additional information is necessary to apportion the error. The paper heavily relies on a previous publication by Fountoukis et al. (2011), who compared the PMCAMx predictions of aerosol composition for the same period with ground based and airborne AMS observations. However, that comparison was fairly limited, including the data from only 4 stations for OA and secondary inorganic aerosols in $PM_1$. The paper is clearly written and the methodology is presented in an understandable way. The authors have adequately answered majority of the previous reviewer requests with a few exceptions.

The screening of the high coarse particle concentrations periods is one of the original characteristics of this work. This allows us to focus on the ability of the model to reproduce the organic and secondary inorganic aerosols as well as their water uptake. The use of one period with high photochemical activity also enables us to test if the model can reproduce the high variability (in space and time) of the corresponding concentrations. The use of detailed AMS measurements from ground stations and an airplane adds more detail to this comparison. In the revised paper we have added the evaluation of the model against additional ground measurements that were available during this period. Detailed responses to the comments of the reviewer can be found below.

**(2)** I agree with the previous reviewers, that more extensive comparison with the observations of e.g. the EMEP network would give higher confidence in the models ability to reproduce the near-surface concentrations of the aerosol components in Europe. Currently no evaluation is presented for instance for the modelled sea salt and the paper does not discuss how it could influence the AOD predictions over the sea.

We have compared the predictions of $PM_{2.5}$ and $PM_1$ of PMCAMx against the corresponding EMEP daily measurements in 27 stations. A total of 795 data points have been used for the evaluation. The model showed very little bias (fractional bias equal to -0.07) and reasonable scatter (fractional error equal to 0.49). The average predicted PM2.5 concentration was 9.07 μg m$^{-3}$ while the average observed was 9.82 μg m$^{-3}$. This performance is quite similar to the one reported by Fountoukis et al. (2011) for the EUCAARI stations and airborne campaign for the same period. A summary of this intercomparison has been added to Section 4 and the detailed results have been added to the Supplementary Information (Tables S4 and S5). Please note that the high sea-salt periods are also screened together with the dust. Therefore this paper does not address the ability of PMCAMx to reproduce the sea-salt concentrations. We have added this point in several points including the abstract and conclusions in the revised manuscript.

**(3)** The temporal and spatial correlations of the model with the MODIS and AERONET observations should also be reported, as was requested by referee #2 - the

current set of model scores does not provide the information on how well the model can reproduce the AOD patterns and variations.

The ability of the model to reproduce the spatial pattern of the measured AODs is shown in Figure 5 of the original paper in which the station-average AODs are compared. The maps of Figure 3 also illustrate this comparison in the form of a map. We have added in the revised paper the corresponding correlation coefficients for completeness. Comparisons of the predicted and measured diurnal average AOD profiles in 24 AERONET stations are shown now in the Supplementary Information. A lot of the diurnal profiles are relatively flat during the period of the AERONET measurements so quantitative correlation analysis is rather misleading in this case.

**(4)** The authors have presented a variety of model scores for a number of regions, however, they have not adequately discussed the discrepancies in the modelled and observed AOD patterns over Europe. What is the reason for underestimating the high AODs, while the low AODs are overestimated? How does the AOD underestimation in sea areas compare with PMCAMx ability to model sea salt?

This is an interesting point that clearly needs additional discussion. Based on the monthly average AERONET observations (see Figure 5b) there is no indication that the model underestimates the high AODs and overestimates the low ones. On average, it does a reasonable job in both. However, when one examines the individual measurements (Figure 6) the range of the measurements exceeds that of the predictions. There are a number of possible reasons for this behavior that is often encountered in chemical transport models. The use of the same anthropogenic emissions inventory every day (with the exception of weekends) is one reason. These emissions do vary from day to day, however the model uses their average missing in the process both ends of the actual air pollution distribution. Measurement uncertainty is a second reason. This will also tend to extend the range of the measured AOD distribution compared to the predicted one. Errors in meteorological fields can also contribute to this discrepancy. Please note that the omission of the high coarse PM periods from the evaluation data set has also eliminated the high sea-salt concentration periods. As a result, the present work offers little insight about the ability (or lack there-of) of PMCAMx to model sea salt (see also our response to Comment 2 above). We have added this discussion to the revised paper.

Specific remarks:

**(5)** Page 4, l 17 - change EUCAARI intensive to EUCAARI intensive campaign.
We have made the correction.

**(6)** Page 5, lines 9-12 "The major new methodological improvement in this effort is the screening of the satellite retrievals for periods with high dust (or coarse particles in general) concentrations as well as the combination of the MODIS and AERONET datasets so that the conclusions can be more robust." - Unclear, please restate.
We have rephrased this complicated sentence.

**(7)** Page 6, lines 22-24. The authors say that they use gas phase anthropogenic emissions from GEMS inventory and carbonaceous aerosols from EUCAARI. However, the anthropogenic emissions also have a non-carbonaceous component, and based on Kuenen et al. (2014) this is not negligible for $PM_{2.5}$ in Europe. Was the noncarbonaceous fraction of primary anthropogenic PM ignored in the model computations?

This is a good point. The non-carbonaceous anthropogenic emissions used in this study have been taken from the GEMS inventory so they are not ignored. This information has been added to the model description section.

**(8)** Page 7, line 6. IS4FIRES provides $PM_{10}$ emissions. How was this speciated to PMCAMx species and sizes?

The size and composition distribution used is based on Andreae and Merlet (Global Biogeochem. Cycles, 15, 955–966, 2001). We assume that approximately 75% of the emissions are in the $PM_{2.5}$ fraction and the rest in the coarse fraction. The average composition (the particles consist mostly organic aerosol and BC) is also based on Andreae and Merlet (2001). This information has been added to the revised paper.

**(9)** Page 8, lines 16-18. How is the aerosol size used in the Mie computations? Is the mean bin diameter used or is there an integration over the size bin?

The mean bin diameter is used for the Mie computations. We have evaluated the accuracy of this simplification by performing detailed calculations over the full diameter range assuming uniform mass or number distributions in each bin. In all cases examined, the differences in the estimated AOD were at most a few percent justifying our simplification. This information has been added to the paper.

**(10)** Page 11, line 2 - What is the explanation, why only 0.4% of the MODIS AODs get discarded over water due to dust influence, while for AERONET this fraction was much higher and very similar to land?

This is mostly due to the fact that a lot of the periods with high dust levels are also accompanied by cloud cover over water. As a result there are no MODIS AOD retrievals during these periods thus lowering the corresponding fraction that needs to be discarded due to dust influence. The location of the AERONET stations also contributes to this discrepancy. A sentence making this point has been added to the manuscript.

**(11)** Page 11, line 23 - Page 12, line 4. Replace the number of data points with the number of stations and flights.

We have added the information about the number of stations (four) and number of flights (sixteen) to the paper. We have kept the number of points to stress that the evaluation was performed based on hourly measurements (for the ground stations) and 5 min averages (for the flights).

**(12)** How does the model bias at ground stations compare with the bias at higher altitudes and how this is expected to influence the modelled AOD?

The model biases for organics and sulfate, the two major $PM_1$ components, were quite similar at the ground and higher altitudes (Fountoukis et al., 2011). For example for organic aerosol the mean bias was -0.4 $\mu g\ m^{-3}$ in both cases while for sulfate it was +0.1 $\mu g\ m^{-3}$ at the ground and -0.1 $\mu g\ m^{-3}$ aloft. The model reproduced well the almost zero ground $PM_1$ nitrate levels in the Eastern Mediterranean (Finokalia) (mean bias 0.02 $\mu g\ m^{-3}$) and the moderate levels in Central Europe (Melpitz) (mean bias -0.1 $\mu g\ m^{-3}$). In the high ammonium nitrate region based on the Cabauw measurements the nitrate bias at the ground was +0.8 $\mu g\ m^{-3}$. The mean bias at higher altitudes was -0.2 $\mu g\ m^{-3}$. This information has been added to the paper. In general, we do not

expect any complications because of this issue given the relatively small biases in most cases.

**(13)** The authors should discuss the discrepancies in the modelled and observed AOD patterns. Currently major differences are visible between the model and MODIS maps on Figure 3, which are not adequately addressed in the paper. What is the reason for underestimating the high AODs in England, Northern Italy, Balkans region and southern part of the Eastern Europe region, while the low AODs are overestimated in Scandinavia, Northern Russia and Central Europe? Does the AOD underestimation in Atlantic, Mediterranean and Black Sea reflect underestimations of sea salt? The temporal and spatial correlations of the model with the MODIS and AERONET observations should also be reported, as was already requested by referee #2.

We have followed the reviewer's suggestion and extended the discussion of the discrepancies between the predicted and observed values. We have included the results of the comparisons of the EMEP measurements in the corresponding discussion. The ability of the model to reproduce the spatial pattern of the measured AODs is shown in Figure 5 of the original paper in which the station-average AODs are compared. The maps of Figure 3 also illustrate this comparison in the form of a map. We have added in the revised paper the corresponding correlation coefficients for completeness. Comparisons of the diurnal average AOD profiles in several AERONET stations are shown in the Supplementary Information.

**(14)** Figure 4 - if possible, redraw with colour scale which would clearly distinguish between the over- and underestimations and accurate predictions (e.g. white around zero).

We have redrawn this figure following the reviewer's suggestion.

**(15)** Page 14, lines 16-17, Figure 5 - are the error envelopes computed for the monthly average values?

We now clarify that the error envelopes shown in Figure 5 are those discussed in the end of Section 3.

**(16)** Page 16, lines 22-24. Fountoukis et al. (2011) reports about as large negative absolute bias for organic aerosol in Finokalia as for sulphates. That should also contribute to the AOD underestimation.

The concentration of sulphate in Finokalia was more than twice that of organic aerosol. Given the difference in hygroscopicity between the two, most of the AOD (more than 80%) is due according to PMCAMx to sulfates. However, the reviewer is right that the organics are probably making a contribution to the AOD underestimation. We have added this point to the discussion.

**(17)** Page 18, line 14. As AOD at all other seas is also underestimated, could sea salt be underestimated in PMCAMx?

This underestimation is clearly possible. Unfortunately the predicted concentrations over the western Atlantic are heavily influenced by the boundary conditions used for the left side of our modeling domain. Any underestimation of these boundary conditions could also explain the small underestimation of the AOD in this area. As a result, we would prefer not to speculate on the accuracy of the sea-salt predictions of the model.

**(18)** Page 18, lines 16-17. Model absolute bias is a better statistic than the fractional bias when explaining the AOD underestimation on monthly level.

We agree with this point and we have added the mean bias (-0.5 µg m$^{-3}$) for sulfate which does help explain the AOD underestimation in this area.

**(19)** Page 19, line 12. Please change to "Sulfates were the major fine PM components predicted in the Black Sea region during the simulation period."

We have rephrased this sentence following the reviewer's suggestion.

**(20)** Page 20, lines 12-14. When increasing the particle size, was the particle number kept constant? I would expect that when mass concentrations would be kept constant, AOD would decrease noticeably.

We now clarify that the number of particles was not kept constant during this sensitivity test. The lack of sensitivity to particle size for typical ambient size distributions is discussed in detail by Pilinis et al. (Sensitivity of direct climate forcing by atmospheric aerosols to aerosol size and composition, J. Geophys. Res., 100, 18739-18754,1995). The reference and a brief discussion have been added to the paper.

**(21)** Page 20, lines 15-18. Usually when discussing the BC mixing state, the BC core is assumed to get coated with a scattering layer, which scatters the solar radiation towards the absorbing core, thus increasing both the particle absorption and scattering. Neither of the cases presented here (internal volume averaged mixture and external mixture) take into account that option.

Our initial test of the sensitivity of the AOD in this area to the BC mixing state considering the two models (external mixture and homogeneous internal mixture) showed that the AOD was not sensitive to this assumption. These represent relative extreme cases. The results are reasonable given the relatively low levels of BC and the high levels of secondary inorganic and organic aerosol during this period. The reviewer is right that one can add the case of the core-shell model to the discussion. However, given the low sensitivity the expected changes will be of the order of 1% or so. We have added this point to the revised paper.

**(22)** Page 21, line 10. I would strongly discourage the authors from using the word "excellent" when describing their results, while discrepancies between the model predictions and observations are clearly visible on Figure 3 for majority of the regions they mention.

We have defined the performance characterization "excellent" in parenthesis as absolute fractional bias less than 15% and fractional error less than 35%. To avoid confusion we have placed the term in quotes. We have also defined "good" and "average" and we believe that under these conditions these terms can be used. Please also note that this performance only applies to specific regions in the domain and not in the regions where major differences exist.

Extra references

Kuenen, J.J.P., Visschedijk, A. J.H., Jozwicka, M., Denier van der Gon, H. A. C., 2014. TNO-MACCII emission inventory: a multi-year (2003-2009) consistent highresolution European emission inventory for air quality modelling. Atmos. Chem. Phys. Discuss. 14, 5837–5869. doi:10.5194/acpd-14-5837-2014.

**Responses to Reviewer #2**

**(1)** In the interests of keeping the peer review process moving forwards, I have decided to submit a review of this paper while also continuing as the Handling Topical Editor. Unfortunately the second anonymous referee for this round of review has not been able to submit their review in a timely manner.

We do appreciate the extra effort by the editor to expedite the review of this manuscript.

**(2)** The manuscript of Panagiotopoulou et al. is currently in its second round of review. The anonymous reviewers of the original submission suggested several revisions, many of which the authors have implemented, and which have improved the paper. As the anonymous reviewer in this second round points out, however, not all of the recommendations from the first round have been implemented in the revised version of the manuscript. The main outstanding points are 1) the limited nature of the AOD comparison, focusing on monthly averages with no assessment of spatial or temporal correlation, and 2) the speculative nature of the explanations offered for model/measurement agreement and discrepancies.

We have done our best to address both of these points as explained below, under the specific comments.

**(3)** Regarding the first point, the addition of a paragraph at the bottom of page 19 is a welcome step towards evaluation of the temporal performance of the model. It is not clear exactly what is shown in Figures S3 and S4 though. I presume that these are average diurnal cycles over the whole month of May 2008, right? If so this should be explicitly stated. In any case this extra analysis falls short of the request of the anonymous reviewer in the first round of review to see a spatial and temporal correlation analysis of the comparison between the simulated and measured AOD. This request has been repeated by the anonymous referee from the second round of review.

We now explicitly state that these are the diurnal averages of the measured and predicted AODs. The ability of the model to reproduce the spatial pattern of the measured AODs is shown in Figure 5 of the original paper in which the station-average AODs are compared. The maps of Figure 3 also illustrate this comparison in the form of a map. We have added in the revised paper the corresponding correlation coefficients for completeness. Additional comparisons of the diurnal average AOD profiles in 24 AERONET stations are now shown in the revised Supplementary Information. Given the fact that a lot of the average profiles are relatively flat the correlation coefficient do not convey much useful information and we believe that the actual figures are a lot more informative about the ability of PMCAMx to reproduce the average diurnal variation (or lack thereof).

**(4)** Regarding the second point, it is not enough to simply state that the model has been evaluated against surface measurements in a previous publication, then leave it at that. Anonymous referees from both rounds of review have called for a more integrated analysis involving comparison of the modelled aerosol composition at the surface with surface measurements from monitoring stations, such as those operated by EMEP. I also think that such an analysis would help alleviate the speculative nature of the discussion of modelled aerosol composition in this paper.

We have compared the predictions of $PM_{2.5}$ and $PM_1$ of PMCAMx against the corresponding EMEP daily measurements in 27 stations. A total of 795 data points have been used for the evaluation. The model showed very little bias (fractional bias equal to -0.07) and reasonable scatter (fractional error equal to 0.49). The average predicted $PM_{2.5}$ concentration was 9.07 µg m$^{-3}$ while the average observed was 9.82 µg m$^{-3}$. This performance is quite similar to the one reported by Fountoukis et al. (2011) for the EUCAARI stations and airborne campaign for the same period. A summary of this intercomparison has been added to Section 4 and the detailed results have been added to the Supplementary Information (Tables S4 and S5). Unfortunately, there are very few additional EMEP fine PM composition measurements during the simulation period (almost all of them have been already used in the evaluation), so we have to restrict the comparison to the numerous fine PM concentration values.

**(5)** This is particularly important for Section 5.2. To start with, when discussing the "major components" (eg. line 17, page 15) of PM, please be explicitly clear in each case whether you are referring to the major components as simulated by the model, or measured, and if this refers to measurements, please give a reference. If your discussion of aerosol composition in each region does not currently include any comparison with measured aerosol composition, please compare your modelled composition with surface measurements wherever they are available.
We have rephrased, when necessary, Section 5.2 clarifying that we are referring to the simulated components. We also mention the corresponding ground measurements of composition and mass in the discussion of each region.

**(6)** There are only two references made in Section 5.2 to the previous evaluations of Fountoukis et al. It would help a great deal if results from these previous evaluations could be integrated better into your analysis. Please describe in the introduction to the paper in more detail what was evaluated and where in these previous studies, and refer back to this explicitly from Section 5.2 when discussing simulated aerosol composition in relevant regions.
We have added details about the previous evaluation in the introduction and also added the corresponding additional information in the discussion of each section.

[revised manuscript text omitted]

---

## Author Response (AR4)

**Response to the Editor's Comments**

In my opinion you have done a good job of answering the comments of the reviewers by adding extra analysis and clarification where necessary.

There is only one comment which I think you should do a better job of answering, and this is comment #22 of reviewer #1. The reviewer has asked you not to use the term "excellent", and you have responded by putting the word in quotation marks. I understand your response, in that you wish to highlight the fact that you have created your own criteria for "excellent", "good", and "average", but I think the way you have done this will lead to more confusion. I recommend the following solution.

At the beginning of Section 5.2, please devote some text to describing each of these categories, along with the rationale for the thresholds of fractional bias and fractional error that you choose. Then, when you discuss each region, you can write something like "Based on these values of fractional bias and fractional area, we classify the performance of PMCAMx in this region as "excellent" (using the criteria given above)". Then, in the conclusions, when you describe the model performance in each region as either "excellent", "good", or "average", you can replace the contents of the following parentheses with the words "based on our own evaluation criteria". I think this will help to make it clear to the readers what you mean here.

We have followed the editor's suggestion and added the definition of the performance levels in the beginning of Section 5.2. These definitions are not ours, but have been proposed by Morris et al. (2005) for regulatory PM modeling in the US and have been often used during the last decade. A reference to Morris et al. (2005) has also been added. When these criteria are used in the rest of the paper we do remind the reader about the source of the definitions and we keep the terms in parenthesis to avoid confusion.

[revised manuscript text omitted]